

# Closing the gap on lower cost air quality monitoring: machine learning calibration models to improve low-cost sensor performance

Naomi Zimmerman[1], Albert A. Presto[1], Sriniwasa P.N. Kumar[1], Jason Gu[2], Aliaksei Hauryliuk[1], Ellis S. Robinson[1], Allen L. Robinson[1], Ramachandran Subramanian[1]

[1]Center for Atmospheric Particle Studies, Carnegie Mellon University, Pittsburgh, 15213, USA
[2]Sensevere LLC, Pittsburgh, 15222, USA

*Correspondence to*: R. Subramanian (subu@cmu.edu)

**Abstract.** Low-cost sensing strategies hold the promise of denser air quality monitoring networks, which could significantly improve our understanding of personal air pollution exposure. Additionally, low-cost air quality sensors could be deployed to

areas where limited monitoring exists. However, low-cost sensors are frequently sensitive to environmental conditions and pollutant cross-sensitivities, which have historically been poorly addressed by laboratory calibrations, limiting their utility for monitoring. In this study, we investigated different calibration models for the Real-time Affordable Multi-Pollutant (RAMP) sensor package, which measures $CO$, $NO_2$, $O_3$, and $CO_2$. We explored three methods: 1) laboratory univariate linear regression, 2) empirical multivariate linear regression and 3) machine-learning based calibration models using random forests (RF).

Calibration models were developed for 19 RAMP monitors using training and testing windows spanning August 2016 through February 2017 in Pittsburgh, PA. The random forest models matched (CO) or significantly outperformed ($NO_2$, $CO_2$, $O_3$) the other calibration models, and their accuracy and precision was robust over time for testing windows of up to 16 weeks. Following calibration, average mean absolute error on the testing dataset from the random forest models was 38 ppb for CO (14% relative error), 10 ppm for $CO_2$ (2% relative error), 3.5 ppb for $NO_2$ (29% relative error) and 3.4 ppb for $O_3$ (15% relative

error), and Pearson r versus the reference monitors exceeded 0.8 for most units. Model performance is explored in detail, including a quantification of model variable importance, accuracy across different concentration ranges, and performance in a range of monitoring contexts including the National Ambient Air Quality Standards (NAAQS), and the US EPA Air Sensors Guidebook recommendations of minimum data quality for personal exposure measurement. A key strength of the RF approach is that it accounts for pollutant cross sensitivities. This highlights the importance of developing multipollutant sensor packages

(as opposed to single pollutant monitors); we determined this is especially critical for $NO_2$ and $CO_2$. The evaluation reveals that only the RF-calibrated sensors meet the US EPA Air Sensors Guidebook recommendations of minimum data quality for personal exposure measurement. We also demonstrate that the RF model calibrated sensors could detect differences in $NO_2$ concentrations between a near-road site and a suburban site less than 1.5 km away. From this study, we conclude that combining RF models with the RAMP monitors appears to be a very promising approach to address the poor performance that

has plagued low cost air quality sensors.



## 1 Introduction

Historically, spatial coverage of air quality monitoring stations has been limited by the high cost of instrumentation; urban areas typically rely on a few reference-grade monitors to assess population scale exposure. However, air pollutant concentrations often exhibit significant spatial variability depending on local sources and features of the built environment (Marshall et al., 2008; Nazelle et al., 2009; Pugh et al., 2012; Tan et al., 2014), which may not be well captured by the existing monitoring networks. In the past several years, there has been a significant increase in the development and applications of low-cost sensor-based air quality monitoring technology (Lewis and Edwards, 2016; McKercher et al., 2017; Moltchanov et al., 2015; Snyder et al., 2013). The use of low-cost air quality sensors for monitoring ambient air pollution could enable much denser air quality monitoring networks at a comparable cost to the existing regime. Increasing the spatial density of air quality monitoring would help quantify and characterize exposure gradients within urban areas and support better epidemiological models. Additionally, more highly resolved air quality information can assist regulators with future policy planning, with identification of hot spots or potential areas of concern (e.g., fracking in rural areas) where more detailed characterization is needed, and with risk mitigation for noncompliant zones. Furthermore, low-cost air quality sensors are generally characterized by their compact size and low power demand. These features enable low-cost sensors to be moved with relative ease to rural areas or developing regions where limited monitoring exists.

A primary challenge with low-cost air quality sensors is calibration at typical ambient pollutant concentrations and environmental conditions. These sensors are prone to cross-sensitivities with other ambient pollutants (Bart et al., 2014; Cross et al., 2017; Masson et al., 2015b; Mead et al., 2013). The most common example is for ozone electrochemical sensors, which also undergo redox reactions in the presence of $NO_2$. Additionally, NO has also been observed to interfere with $NO_2$ and CO sensors have exhibited some cross-sensitivity to molecular hydrogen in urban environments (Mead et al., 2013). Furthermore, low-cost sensors can be affected by meteorology (Levy, 2014; Masson et al., 2015b; Pang et al., 2017; Williams et al., 2013). Most electrochemical sensors are configured such that the reactions are diffusion-limited, and the diffusion coefficient can be affected by temperature (Hitchman et al., 1997); Masson et al. (2015b) have shown that at relative humidity exceeding 75% there is significant error, possibly due to condensation on potentiostat electronics. Lastly, the stability of low-cost sensors is known to degrade over time (Jiao et al., 2016; Masson et al., 2015a). For example, in electrochemical cells, the reagents are consumed over time and have a typical lifetime of 1-2 years.

Deconvolving the effects of cross-sensitivity and stability on sensor performance is complex. Linear calibration models developed in the laboratory perform poorly on ambient data (Castell et al., 2017). Attempts to build calibration models from first principles have shown some success, but the models are difficult to construct and their transferability to new environments remains unknown (Masson et al., 2015b). Accurate and precise calibration models are particularly critical to the success of dense sensor networks deployed in urban areas of developed countries where concentrations are on the low end of the spectrum



of global pollutant concentrations, as poor signal-to-noise ratios may hamper their ability to distinguish between intra-urban sites. As such, there has been increasing interest in more sophisticated algorithms (e.g., machine learning) for low cost sensor calibration. To date, there have been published studies using high-dimensional multi-response models (Cross et al., 2017) and neural networks (Esposito et al., 2016; Spinelle et al., 2015, 2017). Spinelle et al. (2015) showed that artificial neural network

calibration models could meet European data quality objectives for measuring ozone (uncertainty < 18 ppb); however, meeting these objectives for $NO_2$ remained a challenge. Cross et al. (2017) built high-dimensional multi-response calibration models for CO, NO, $NO_2$ and $O_3$ which had good agreement with reference monitors (slopes 0.6-0.96, $R^2$ 0.51-0.96). Esposito et al. (2016) demonstrated excellent performance with dynamic neural network calibrations of $NO_2$ sensors (mean absolute error < 2 ppb); however, the same performance for $O_3$ was not observed. Furthermore, these calibrations have only been tested on a

small number of sensor packages. For example, Cross et al. (2017) tested two sensor packages, each containing one sensor per pollutant over a four-month period, of which 35% was used as training data. Spinelle et al. (2015) tested a cluster of sensors in a single enclosure, testing 22 individual sensors in total over a period of 5 months, of which 15% was used as training data. Esposito et al. (2016) reported calibration performance on a single sensor package (5 gas sensors per package for measuring NO, $NO_2$ and $O_3$) and the model was tested on four weeks of data.

In this study, we aim to improve the calibration strategies of low-cost sensors using a random-forest-based machine learning algorithm, which, to our knowledge, has not been previously applied to low-cost air quality monitor calibrations. To ensure calibration model robustness, they were developed and validated for 19 sensor packages, with each package containing one sensor per species (CO, $CO_2$, $NO_2$, $SO_2$ and $O_3$) for a total of 95 individual sensors. Furthermore, the model training and testing

was conducted over a six-month period (August 2016 – February 2017) spanning multiple seasons and a wide range of meteorological conditions, providing one of the most comprehensive low-cost air quality sensor calibration investigations to date. The fitting of the machine learning algorithms is discussed in detail to determine ideal calibration datasets to maximize performance and minimize overtraining. The performance of the random forest models is compared to traditional laboratory univariate linear models, multiple linear regression models, and EPA performance guidelines. The performance of a given

model over time is also discussed.

## 2 Experimental methods

### 2.1 Measurement site

Measurements were made from August 3, 2016 to February 7, 2017 on the Carnegie Mellon University campus in the Oakland neighbourhood of Pittsburgh, PA. The measurement site (40°26'31.5"N, 79°56'33"W) is located within small (< 100 vehicles)

limited access, open air parking lot near the center of campus. It consisted of a mobile laboratory equipped with reference-grade instrumentation (Section 2.3) and adjacent lawn space where the RAMP monitors were mounted on tripods (Section 2.2). The dominant local source at the site is vehicle emissions when vehicles enter and exit the parking lot during the morning





and evening rush hours. The small size of the parking lot (< 100 cars) and few other local sources means that for most of the day the location is essentially an urban background site. During the measurement period, the site mean (range) ambient temperature and relative humidity were 13°C (-15 to 34 °C) and 71% (27 to 98%), respectively.

## 2.2 Real-time Affordable Multi-Pollutant (RAMP) monitor

The study uses the Real-time Affordable Multi-Pollutant (RAMP) monitor, which was developed in a collaboration between Carnegie Mellon University and SenSevere. The RAMP monitor incorporates widely-used Alphasense electrochemical sensors to measure gaseous pollutants (CO, $NO_2$, $SO_2$ $O_3$) and a non-dispersive infrared (NDIR) sensor to measure $CO_2$. The latter sensor also includes modules to measure temperature and relative humidity. The RAMP is paired with a Met-One Neighborhood PM monitor to measure optical $PM_{2.5}$. The RAMP uses the following commercially-available electrochemical

sensors from Alphasense Ltd: carbon monoxide (CO, Alphasense ID: CO-B41), nitrogen dioxide ($NO_2$, Alphasense ID: NO2-B43F), sulfur dioxide ($SO_2$, Alphasense ID: SO2-B4), and total oxidants ($O_x$, Alphasense ID: Ox-B431). The unit also includes a nondispersive infrared (NDIR) $CO_2$ sensor (SST CO2S-A) which contains built-in T (method: bandgap) and RH (method: capacitive) measurement. The experiments involved 95 individual pollutant sensors mounted in 19 unique RAMP monitors.

The electrochemical sensor outputs were measured using electronic circuitry custom designed by SenSevere optimized for signal stability. The circuitry includes custom electronics to drive the device, multiple stages of filtering circuitry for specific noise signatures, and an analog-to-digital converter for measurement of the conditioned signal. The RAMP monitors are housed in a NEMA-rated weather proof enclosure (Figure 1A) and equipped with GSM cards to transmit data using cellular networks to an online server. The RAMP monitors also log data to an SD card as a fail-safe in case of wireless data transfer issues. The

sensors sample passively from the bottom of the unit (Figure 1B), with screens installed to protect the sensors. If operated with the $PM_{2.5}$ monitor, the RAMP monitors require 120-240V AC power; however, roughly 3 weeks of measurements of gaseous species, T, and RH are possible on single charge of a built-in 30 amp-hour NiMH battery. The RAMP monitors are either mounted to a steel plate for easy pole mounting or are deployed on tripods approximately 1.5 m above the ground (Figure 1C). In this study, all the RAMP monitors were tripod-mounted at a consistent height.

In their simplest configuration, electrochemical sensors function based on a redox reaction within an electrochemical cell in which the target analyte oxidizes the anode and the cathode is proportionally reduced (or vice versa, depending on target analyte). The subsequent movement of charge between the electrodes produces a current which is proportional to the analyte reaction rate, which can be used to determine the analyte concentration. The Alphasense electrochemical sensors utilize a more

complex configuration by using four electrodes (working, reference, counter and auxiliary) to account for zero current changes. Essentially, the auxiliary electrode, which is not exposed to the target analyte, accounts for baseline changes in the sensor baseline signal under different meteorological conditions. Additional details on the theory of operation for electrochemical sensors can be found in Mead et al. (2013).



The RAMP monitors log two output signals from each of the Alphasense sensors: one from the auxiliary electrode and the other from the working electrode. The net sensor response is determined by subtracting the auxiliary electrode signal from that of the working electrode. In theory, for a target analyte a linear relationship should exist between the net sensor signal for that

analyte and ambient analyte concentrations, and this expectation forms the basis of univariate linear regression models built from laboratory calibrations. However, as noted in the introduction, even with an auxiliary electrode, electrochemical sensors may insufficiently account for the impacts of temperature (which affects the rate of diffusion) and relative humidity under high humidity conditions where condensation is possible. This has motivated researchers to construct multivariate linear regression models (MLR) to account for these temperature and humidity effects (Jiao et al., 2016). While these calibration models

typically improve performance relative to univariate linear models (Spinelle et al., 2015, 2017), they typically do not incorporate any cross-sensitivities to other pollutants or any non-linearities in the response. In this study, we attempt to build a calibration model for each analyte with no underlying assumptions regarding the calibration model structure and allow the models to consider directly the full suite of data being reported by the RAMP monitors using a machine learning approach.

## 2.3 Reference instrumentation

Reference measurements were made on ambient air continuously drawn through an inlet on the roof of the supersite located approximately 2.5 m above ground.  Gaseous pollutants were drawn through approximately 4 m of 0.953 cm outer diameter Teflon fluorinated ethylene propylene (FEP) tubing with a six-port stainless steel manifold for flow distribution to the gas analyzers. Measurements were made using direct absorbance at 405 nm for $NO_2$ (2B Technologies Model 405 nm), a gas filter correlation infrared analyzer for CO (Teledyne T300U), a non-dispersive infrared analyzer for $CO_2$ (LICOR 820), UV

absorption for $O_3$ (Teledyne T400 Photometric Ozone Analyzer) and by UV fluorescence for $SO_2$ (Teledyne T100A UV Fluorescence $SO_2$ Analyzer). The time resolution for all reference measurements was 1 s.

The reference gas analyzers were checked and calibrated weekly using calibration gas mixtures, except for $O_3$ which is calibrated biannually at a nearby regulatory monitoring site. The CO and $NO_2$ analyzers experience modest baseline drift between weekly calibrations, on the order of approximately 40 ppb for CO and 2 ppb for $NO_2$. Hence, baseline pollutant

concentrations were normalized to a nearby regulatory monitoring site (Allegheny County Health Department, Air Quality Division, Pittsburgh, PA). The gas analyzers at the regulatory monitoring site are checked daily and thus this normalization helped correct for any baseline drift during the days between calibration. No significant drift was observed for $CO_2$ or $O_3$.

## 3 Calibration methods

Three calibration methods were evaluated: (1) a laboratory-based univariate linear regression based on net sensor response when exposed to calibration gases, (2): an empirical multivariate linear regression of net sensor response, T and RH regressed





against reference monitor concentrations, and (3): a random forest machine learning model using net responses from all sensors, T, and RH to predict reference monitor concentrations. Calibration models were constructed for the CO, $NO_2$, $CO_2$ and $O_3$ sensors in each RAMP monitor. In this study, no calibration models were built for $SO_2$ due to $SO_2$ concentrations measured with the reference instrumentation being below the instrument detection limit (<0.4 ppbv) for most of the campaign

(no nearby sources of $SO_2$). While lab calibrations were conducted for the $SO_2$ sensors, this data will be the subject of a future publication on air quality in industrial areas where $SO_2$ is more commonly detected.

### 3.1 Laboratory-based univariate linear regression (LAB)

Prior to outdoor collocation, the sensors inside the RAMP monitors were calibrated in a laboratory environment using a custom manufactured sensor bed and calibration gas mixtures. The sensors were exposed to each step in the calibration window (Table

1) for 20 minutes and a flow rate of 9 LPM flowed perpendicular to the sensor surface. The sensor response at each calibration step was averaged once the signal had stabilized. Temperature and relative humidity were not controlled during the calibration. The temperature was at levels typical of indoor laboratory environments (approx. 20 °C), and the dry calibration gas provided very little humidity (RH <5%). Calibrations were built for CO, $NO_2$ and $CO_2$. Laboratory calibrations for $O_3$ were not performed.

The laboratory calibration follows a standard univariate linear regression model of regression net (CO, $NO_2$) or raw ($CO_2$) signal against the reference gas concentration (Eq. 1)

$$\text{Corrected}_{\text{Lab Cal}} = \beta_0 + \beta_1 \times [\text{Net Sensor Response (CO, NO}_2\text{) or Raw Sensor Response (CO}_2\text{)}], \qquad (1)$$

Model performance was evaluated by comparing the calibrated response to reference measurements. We refer to the laboratory univariate linear regression calibration as LAB. Separate LAB calibrations were developed for each sensor (95 individual calibrations).

### 3.2 Empirical multivariate linear regression (MLR)

Following laboratory calibration, the individual sensors were mounted in the RAMP monitors and deployed adjacent to the Carnegie Mellon University supersite. The collocation period varied by RAMP, with a minimum collocation period of 6 weeks and a maximum collocation period of the entire 6-month study period. The collocation window varied due to intermittent deployment of some RAMP monitors for ongoing air quality monitoring campaigns in the Pittsburgh area. To build calibration models, the collocation period was separated into a training and testing period identical to that used for the random forest

calibration (see Section 3.3). Due to the previously established influence of T and RH on sensor response (Jiao et al., 2016; Masson et al., 2015b; Spinelle et al., 2015, 2017), a multiple linear regression (MLR) model was used to calibrate the output from each sensor using net sensor response to the target analyte (e.g. CO for the CO sensor), T and RH as explanatory variables



(Eq. 2), similar to the approach described in a recent a European Union report on protocols for evaluating and calibrating low-cost sensors (Spinelle et al., 2013).

$$Corrected_{MLR} = \beta_0 + \beta_1 \times [\text{Net Sensor Resp. (CO, NO}_2\text{, O}_3\text{) or Raw Sensor Resp. (CO}_2\text{)}] + \beta_2 \times T + \beta_3 \times RH, \qquad (2)$$

The training data was used to calculate the model coefficients ($\beta_0$ through $\beta_4$) and the model performance was evaluated on withheld testing data. Separate multivariate linear regression models were developed for each sensor (95 individual models). We refer to these models as MLR.

### 3.3 Random forest model (RF)

A random forest (RF) model is a machine learning algorithm for solving regression or classification problems (Breiman, 2001). It works by constructing an ensemble of decision trees using a training data set; the mean value from that ensemble of decision trees is then used to predict the value for new input data. Briefly, to develop a random forest model, the user specifies the maximum number of trees that make up the forest, and each tree is constructed using a bootstrapped random sample from the training data set. The origin node of the decision tree is split into sub-nodes by considering a random subset of the possible explanatory variables. The training algorithm splits the tree based on which of the random subsets of explanatory variables is

the strongest predictor of the response. The number of random explanatory variables considered at each node (denoted $m_{try}$) is tuned by the user. This process of node splitting is repeated until a terminal node is reached; the user can specify the maximum number of sub-nodes or the minimum number of data points in the node as the indication to terminate the tree. For our random forest models, the terminal node was specified using a minimum node size of 5 data points per node.

To illustrate the method, consider building a random forest model for one RAMP monitor using a single decision tree and a subset of 100 training data points to build a CO calibration model (Figure 2). In this highly simplified example, at the first node, the net CO sensor signal is the strongest predictor of the CO reference monitor concentration, with a natural split in the data at a net CO sensor voltage of 255.9 a.u. If sensor voltage exceeds 255.9, a cluster of 7 data points from the training data

predicts an average CO concentration of 357 ppb, if CO net sensor voltage is ≤255.9 then the data goes to the next decision node, in which net CO sensor signal is again the strongest predictor of the CO reference monitor concentration, with a natural break in the data at a net CO sensor voltage of 167.3 a.u. The splitting proceeds until all the training data are assigned to a terminal node. The prediction value for each terminal node is the average reference monitor concentration of training points assigned to that node. To apply the algorithm (i.e. predict the CO concentration from a set of measured inputs), the user takes

the measured T and the net CO, NO$_2$ and O$_3$ signals and follows the path through the tree to the appropriate terminal node. The predicted CO concentration for that tree is then the average training value associated with that terminal node. This process is then repeated through multiple trees (Figure 2 shows only one simple tree) and the predictions from each tree are averaged to determine the final output from the entire random forest model. In this simple example, there are only six possible CO



concentrations the random forest model will output. In practice, each tree has hundreds of terminal nodes and the forest typically comprises hundreds of trees, which means that there are thousands of possible answers. The model prediction for a given set of inputs is the average prediction across all the hundreds of trees that comprise the forest.

The random forest model's main limitation is that its ability to predict new outcomes is limited to the range of the training data set; in other words, it will not predict data with variable parameters outside the training range. Therefore, a larger and more variable training data set should create a better final model. To maximize utilization of the training data set to avoid missing any spikes during the training window, a k-fold cross validation approach was used. A k-fold cross-validation divides the data into k equal sized groups (where k is specified by the user) and k repeats are used to tune the model. Consider an example

where k is equal to 5 (a 5-fold cross-validated random forest model). With a 5-fold validation, five unique random forest models are constructed, one for each fold. In building the first random forest, the first 20% (1/k) of the data will be the testing data, and the remaining 80% [(1-k)/k] of the data will be used as training. In building the second random forest, the next 20% of the data will be used as test data, and the first 20% and remaining 60% will be used to train. This is repeated until the data are fully covered, at which point the random forest model is created by combining the five (k) individual models into one large

random forest model. This helps to minimize bias in training data selection when predicting new data, and ensures that every point in the training window is used to build the model.

    In this study, reference gas data, RAMP net sensor data for CO, $NO_2$, $SO_2$, $O_3$, and RAMP raw sensor data for $CO_2$, T, and RH were collected at 15 second resolution, time-matched, and down-averaged to 15 min intervals (IGOR Pro v6.34), which is

higher temporal than the 1 h intervals at which typical regulatory monitoring information are reported. The down-sampled data were then imported into R (ver. 3.3.3, "Another Canoe") for random forest model building. R is an open-source package for tuning and cross-validating many classes of statistical models, including random forest models. The cross-validated random forest models were compiled using the open-source "caret" package (Kuhn, 2017). The model considered all RAMP data (net voltage outputs from the five gas sensors plus T and RH, 7 possible variables total) as potential explanatory variables to predict

the reference monitor gas concentration. The number of trees was capped at 100 per fold, and a five-fold cross-validation was used for a total of 500 trees. Therefore, the predicted value for a given set of measured inputs is the average value from this set of 500 trees (each tree provides one prediction). When fitting the random forest models with the training data, the main tuning parameter is the number of explanatory variables to consider at each decision node ($m_{try}$). To determine the optimal $m_{try}$, the root mean square error (RMSE, equation in Supplemental Information) and the coefficient of determination ($R^2$) were

calculated on the withheld folds of the training data (Figure 3, step 2) for $m_{try}$ equal to 2, 4 or 7 to span the complete variable range. The random subset of explanatory variables considered at each node was chosen based on which value of $m_{try}$ minimized RMSE. The cross-validation and the subset of explanatory variables randomly considered at each node ($m_{try}$) was tuned using the caret package in R (Kuhn, 2017). Following random forest model generation and tuning, the five 100 tree models were combined to create a final model with 500 trees. This process was repeated for each sensor to create 95 separate random forest



models. The final models convert the RAMP output signals into calibrated concentrations. The model conversion was done within R, where it exists as a standalone object compatible with the standard R configuration.

Data from three RAMP monitors (15 individual gas sensors) were used to investigate the optimal training period, which was
determined by comparing the training data size to mean absolute error (MAE, the average of the absolute value of the residuals). The optimal training period was the period beyond which increases in the length of the training window (and therefore size of the training dateset) no longer resulted in significant reductions in the MAE. The initial training window evaluated was 1 week, and 1 week increments in training period duration were considered until MAE was minimized. The optimal collocation window was determined to be 4 weeks (or 2688 data points at 15-minute resolution). This was evaluated
for a consecutive collocation window and for 8 collocation windows equally distributed throughout the whole collocation period (August 2016 – February 2017) in half week increments. Details of this evaluation are provided in the Supplemental Information, but the intermittently distributed collocations generally performed slightly better, with reductions in MAE of 12 ppb (4% relative error) for CO, 2 ppm for $CO_2$ (0.4% relative error), 0.4 ppb for $NO_2$ (4% relative error), and 1.6 ppb for $O_3$ (7% relative error) compared to the consecutive four-week collocation.  The motivation for exploring intermittent collocation
windows dispersed throughout the study period was to ensure that the training period covered a complete range of gas species concentrations, temperatures and relative humidity. In practice, the degree of collocation utilized in this study is equivalent to collocating the RAMP monitors with reference monitors for 3-4 days every 1-2 months. However, if the MAE using the initial consecutive collocation is satisfactory for the application, this calibration strategy was not substantially less accurate than the distributed collocations.

**3.4 Metrics for performance evaluation**

The evaluation of the different models was conducted on 15-minute averaged testing data (i.e., data withheld entirely from model building). Metrics to quantitatively compare the LAB, MLR and RF model output to the reference monitor concentrations included Pearson r, which is a measure of the strength and direction of a linear relationship, and the coefficient of variation of the mean absolute error (CvMAE, Eq. 3). For comparing the RF model performance to other published studies,
we also evaluated mean bias error, mean absolute error, slope of the linear regression of RF model calibrated RAMP data and reference data, and coefficient of determination ($R^2$).

$$\text{CvMAE} = \frac{\text{MAE}}{\text{Avg. Reference Conc.}} = \frac{1}{\text{Avg. Reference Conc.}} \times \left[\frac{1}{n}\sum_{i=1}^{n}|\text{Model}_i - \text{Reference}_i|\right], \tag{3}$$

Another useful tool for visually comparing competing models is a target diagram (Jolliff et al., 2009). A target diagram
illustrates the contributions of the centered root mean square error (CRMSE, which is RMSE corrected for bias) and the mean bias error (MBE) towards total RMSE. In a target diagram, the x-axis is the CRMSE, the y-axis is the MBE and the vector distance to the origin is the RMSE. Since CRMSE is always positive, a further dimension is added: if the standard deviation



of the model exceeds the standard deviation of the measurements, the CRMSE is plotted in the right quadrants and vice versa. To match previously constructed target diagrams (Borrego et al., 2016; Spinelle et al., 2015, 2017), the CRMSE and MBE were normalized by the standard deviation of the reference measurements, and thus the vector distance in our diagrams is RMSE/$\sigma_{reference}$ (nRMSE). The resulting diagram enables visualization of four diagnostic measures: (1) whether the model

tends to overestimate (MBE > 0) or underestimate (MBE < 0), (2) whether the standard deviation of the model is larger (right plane) or smaller (left plane) than the standard deviation of the measurements, (3) whether the variance of the residuals is smaller than the variance of the reference measurements (inside circle of radius 1) or larger than the variance of the reference measurements (outside circle), and (4) the error (nRMSE), the vector distance between the coordinate and the origin. Details of equations required to build a target diagram are provided in the Supplemental Information. Model performance metrics were

calculated in R (ver. 3.3.3, "Another Canoe") using the "tdr" package (Perpinan Lamigueiro, 2015).

## 4 Results and Discussion

### 4.1 Calibration model goodness of fit: comparing model predictions to training data

Following model building, the goodness of fit between the model output concentrations and the reference monitor concentrations during the training window (i.e. the data used to build the model) were evaluated for all three calibration model

approaches (laboratory univariate linear regression "LAB", field-based multiple linear regression "MLR" and field-based random forest "RF"). For the training period, the calibrated CO and $O_3$ concentrations were all highly correlated (Pearson r > 0.8) with the reference monitor concentrations for all the calibration model approaches (Table 2). However, only the RF model achieved strong correlations between the reference monitor and the RAMPs for $NO_2$ and $CO_2$. Furthermore, CvMAE for each species was ≤5% during the training window for the RF models, substantially outperforming the other models.

Regression plots for all 19 RAMPs and all four gas species illustrating the goodness of fit of the RF model are provided in the Supplemental Information. For the RF models, Table 2 also provides the random subset of explanatory variables sampled for splitting at each decision node ($m_{try}$) to achieve the lowest model RMSE. In general, the larger the $m_{try}$, the simpler the underlying structure of the model. The advantage of a lower $m_{try}$ is that subtle relationships between explanatory variables and

the response can be probed. For example, if there is one dominant variable but the model is permitted to consider all 7 explanatory variables at each decision node, then the model will most frequently split the data based on the dominant variable, potentially masking the effect of other variables on the response. If the goodness of fit of the calibration model is improved by decreasing $m_{try}$, this suggests more complex variable interactions (Strobl et al., 2008).

Using the $m_{try}$ metric, we observed that the underlying RF model structure is the simplest for CO, that some model explanatory variable complexities exist for the $O_3$ and $NO_2$ models, and that the $CO_2$ model is the most complex and relies on subtle relationships between the explanatory variables to best fit the data (lowest $m_{try}$ had the best results). This finding matches our





expectations based on the LAB and MLR models; these simpler models performed best for CO and worst for $CO_2$. The trends in the $m_{try}$ metric highlights the value of the RF model approach which directly accounts for multiple pollutants. This appears to be critical for $O_3$, $NO_2$ and $CO_2$ sensors because they are cross-sensitive to other pollutants. Cross-sensitivities have been shown to have a minimal impact on CO sensors, with the only notable cross-sensitivity being to molecular hydrogen (Mead et

al., 2013). The poor performance of linear models at predicting $CO_2$ concentration is not surprising, as the sensor was observed to measure high concentrations under periods of high relative humidity (e.g., during rain) and in some cases during heavy rain will be saturated at 2000 ppm, the upper limit of the sensor, and then is reset to 400 ppm daily, as per manufacturer recommendations. The increase in $CO_2$ under high humidity conditions is likely due to the interference of water with $CO_2$ in the NDIR signal. Linear models are poorly suited to describe this behaviour.

**4.2 Evaluation of models using testing data**

To test the performance of the three different calibration models, the models were applied to the testing data that were not used for model fitting. The RAMP monitor concentrations after correction using the calibration models were compared to the actual measured reference concentrations (Figure 2, step 5). To illustrate the approach, in Figure 4, we show a very short time-series of the testing data (~48-hour window) for RAMP #1. This RAMP monitor's performance is representative of the average

model performance across the 19 RAMP monitors and therefore illustrates the quality of an average model. Figure 4 also shows the calibrated RAMP #1 output regressed against the reference monitor concentration for the entire testing period for all three calibration models (LAB, MLR, and RF). For this period, the RF model clearly outperformed the LAB and MLR models. Differences between the different models were smallest for CO and $O_3$ and largest for $CO_2$ and $NO_2$; the LAB models essentially did not reproduce the reference concentrations for $CO_2$ and $NO_2$. To illustrate the consistency of the RF model-

calibrated RAMP monitors across the entire suite of monitors, regressions for all the RAMP monitors for $O_3$ are shown in Figure 5. Regression plots for all RAMPs across the other gases are provided in the Supplemental Information.

To assess the overall model performance, two performance metrics (Pearson r and CvMAE) were calcualted for each RAMP monitor using the entire testing dataset (Figure 6). The size of the testing dataset varied from 1.4 to 15 weeks, with a median

value of 5 weeks. This aggregate assessment shows that the MLR and RF models are interchangable for CO, as both models achieved Pearson r >0.9 and CvMAE <15%. The LAB model achieved a similar Pearson r, but CvMAE doubled to ~30%. For $CO_2$, $NO_2$, and $O_3$, the RF model substantially outperforms the LAB and MLR calibration models on the testing data. On average, Pearson r exceeded 0.8 for the RF model for $CO_2$ and $NO_2$ versus < 0.6 for the LAB and MLR calibration models.

Furthermore, the RF model performance was more consistent across the RAMP monitors than the MLR and LAB models. For example, the Pearson r for $NO_2$ ranged from 0.92 to 0.95 for the RF models versus 0.74 to 0.89 for the MLR models. This means that essentially all the RF models for $O_3$ performed well versus only a subset of the MLR models. The consistency of the different models is indicated by the smaller range in the box plots of Figure 6.





To compare the LAB, MLR and RF models, target diagrams were constructed for the four gases using all three calibration models for each RAMP (Figure 7). The target diagrams show that, on average, across the RAMP monitors the random sensor error (distance to origin) was smaller for RF models and the RF models showed the least RAMP-to-RAMP variability (less disperse). This contrasts with the MLR models, whose bias and extent of model standard deviation varied much more widely between RAMPs, especially for $CO_2$. For the LAB models, the error for $CO_2$ and $NO_2$ was approximately an order of magnitude larger than for the RF and MLR models and had to be plotted on a separate inset due to their poor performance. Across all gases, the RF models on average were biased slightly lower than the reference. Thus, we conclude that the low CvMAE, high Pearson r correlations, lowest bias and lowest absolute error characteristics of the RF models for all four gases are significant improvements compared to conventional calibration approaches (LAB and MLR).

### 4.3 Detailed assessment of RF model performance

To investigate the performance of the RF models in greater detail, we assessed the effect of amount of testing data on model performance, the relative importance of the seven explanatory variables, the performance of the models across the different concentration ranges, and the number of data points needed in each concentration range to optimize the fit.

### 4.3.1    Drift over amount of testing data

The first assessment was of amount of testing data. In this study, any data remaining after training were used to test model performance, provided there were at least 48 hours of testing data (192 data points). Again, all the data have 15 min temporal resolution. The number of points used to test the model performance varied by RAMP monitor and by pollutant, as reference monitors were occasionally offline for maintenance and calibration, and some RAMP monitors were intermittently deployed for concurrent air quality monitoring campaigns in Pittsburgh. To assess the effect of number of testing points on conclusions regarding RF model performance, we compared the MAE to the number of points in the testing window (Figure 8). For all the gas species, the MAE was essentially flat across the RAMP monitors; RAMP monitors with more testing data did not have substantially higher (worse) MAE, suggesting the RF models are robust over time. For $NO_2$, the most data available for testing was approximately 8 weeks due to instrument maintenance and repair taking the $NO_2$ reference monitor offline for 6 weeks of the study. Figure 8 also shows MAE over time from one RAMP, RAMP #4, which remained at the Carnegie Mellon supersite for the entirety of the six-month study. MAE was calculated for an increasing cumulative number of weeks forward in time, and again, MAE was consistent (and in some weeks improved) over time.

### 4.3.2    RF model explanatory variable importance

While RF models are non-parametric, some sense of the model structure can be gained by examining the relative importance of the explanatory variables. The importance of each variable was quantified by comparing the percent increase in mean square error (MSE) if the explanatory variable signal is permuted (randomly shuffled). If an explanatory variable strongly affects the





model performance, permuting that variable results in a large increase in MSE. Conversely, if a variable is not a strong predictor of the response, then permuting the variable does not significantly increase the MSE. Figure 9 shows for each of the gases (CO, $CO_2$, $NO_2$ and $O_3$) the increase in MSE when the explanatory variables were permuted. For both CO and $O_3$, the signal from the sensor measuring the target analyte (CO or $O_3$) is the most important explanatory variable, as expected. For the $O_3$,

the second most important variable was the $NO_2$ signal, an expected cross-sensitivity, as the ozone sensor measures total oxidants ($O_3 + NO_2$) (Spinelle et al., 2015).

The explanatory variable importance is more complex for $CO_2$ and $NO_2$. For $CO_2$, all variables are roughly equally important, with CO being the most important. This is likely due to the strong meteorological effect of humidity on the measured $CO_2$

concentration; the model must rely on other primary pollutants to predict the $CO_2$ signal when the measured $CO_2$ has reached full-scale, and short-term fluctuations of $CO_2$ are likely from combustion sources (e.g., vehicular traffic in urban areas) which also emit CO. This highlights the value of having sensors for multiple pollutants in the same monitor. Including measurements of additional pollutants helps the RF model correct for cross-sensitivities. For the $NO_2$ model, RH was the most important explanatory variable followed by the $NO_2$ sensor signal, highlighting again the importance of including meteorological data

within sensor packages. The $NO_2$ model was also more strongly affected by temperature than the other pollutants. We hypothesize that the sensitivity of the $NO_2$ sensor to ambient $NO_2$ is suppressed in Pittsburgh, which has low ambient $NO_2$ concentrations compared to other cities where these sensors have been evaluated (see Table 3). $NO_2$ is lowest when ozone is highest in the summer, and thus the $NO_2$ RF model effectively uses T and RH as indicators for seasonality when $NO_2$ is low and the sensor response is supressed. Furthermore, the relatively equal variable importance of several of the explanatory

variables within a model suggests that a cluster of sensors measuring many different species is critically important to build robust calibration models. The only sensor channel that did not contribute significantly to any model performance was the $SO_2$ sensor, thus this sensor could be replaced with a more relevant sensor, such as NO, in future iterations of the RAMP monitor. These findings highlight the value of bundling sensors for measuring a suite of pollutants together, as the different sensors can capture (at least to some extent) cross-sensitivities to other pollutants and improve the model performance for other sensors.

### 4.3.2    RF model performance as a function of ambient concentration

In Section 4.2, predicted concentrations were normalized to average reference monitor concentration to compare quantitatively differences between the different calibration models (CvMAE). To evaluate the RF model performance at different reference concentrations, the testing data were divided into deciles for which the median reference monitor concentration, the absolute

residual, and the residual normalized to the reference monitor concentration were calculated (Figure 10). For all species, the RF models tended to overestimate at lower concentrations, and underestimate at the highest concentrations. For the CO RF model, the normalized residual is within 10% of the reference monitor concentration by the 20[th] percentile of the data (>100 ppb), and continues to improve until the 50[th] percentile when it plateaus at a normalized residual of about 5%. The US EPA





requires a limit of detection of 100 ppb for CO instruments used for regulatory monitoring (United States Environmental Protection Agency, 2014), thus our performance meets that goal. In the top decile, the average absolute CO residual for the RF models approximately doubles but the relative error is still around 5%. However, the top decile spans the broadest concentration range due to the lognormal shape of the CO concentration distribution, and these points are difficult to capture

in training data sets.

For the $CO_2$ RF model, agreement with the reference monitor data are within a few percent up to the $90^{th}$ percentile, when agreement drops to within 5%. This is possibly due to the RF model actively supressing high $CO_2$ sensor signals, as the sensor is prone to reading erroneously high concentrations during rain events. Additionally, the top decile of the data spans a wide

range of $CO_2$ concentrations due to the lognormal shape of the $CO_2$ distribution. As with CO, the $NO_2$ RF model agreement with the reference monitor plateaus around the $50^{th}$ percentile mark; however, the $NO_2$ RF-model error exceeds 100% for the lowest decile (<5 ppb), suggesting an effective sensitivity of the sensor of 5 ppb. For the $O_3$ RF model, the effective sensitivity is also around 5 ppb; when the average reference monitor concentration increased from 5 ppb to 10 ppb (from first to second decile), the normalized residual decreased from over 100% to about than 20%. The US EPA limit of detection for federal

regulatory monitors is 10 ppb for both $NO_2$ and $O_3$, suggesting that as with CO, the RF model performance is within 20% of regulatory standards (United States Environmental Protection Agency, 2014).

Systematic underprediction at the highest concentrations was also observed and is a consequence of the training dataset used to fit the RF model. Unless the range of concentrations in the training data encompasses the range of concentrations during

model testing, there will be underpredictions for concentrations in exceedance of the training range. Additionally, the performance of the RF model is sensitive to the number of data points at a given concentration and the model performance. To build a robust model, many data points are required at a given concentration to probe the extent of the ambient air pollutant matrix. In this study, the training windows were dispersed throughout the collocation period to ensure good agreement of gas species and meteorological conditions during both the training and testing windows (see Supplemental Information).

To illustrate the impact of number of data points, we binned the data for the representative RAMP (RAMP #1) by concentration and the average concentration measured by the reference monitors was plotted against the average concentration from the calibrated RAMP (Figure 11). The uncertainty in the random forest model was plotted as the standard deviation of the model solutions from the 500 trees and the bins were colour coded by the number of data points within each bin. Figure 11 illustrates

that for every pollutant agreement with the reference monitor and uncertainty in the model prediction was larger for concentration bins containing fewer than 10 data points. This disproportionately impacted the upper end of the pollutant distribution where fewer data points were collected due to the intermittent and variable nature of high pollutant episodes. This suggests that a minimum of 10 data points at a given concentration are needed to adequately train the RF model, which may inform future RF model building. At $NO_2$ concentrations below 5 ppb, deviations from the 1:1 line were also observed despite





the training dataset containing more than 100 data points at these concentrations. As was concluded from Figure 10, 5 ppbv appears to be the sensitivity limit of these low-cost sensors for $NO_2$.

## 4.4    Comparison of results to other published studies

In this section, we compare the performance of our RF models to results from other recent studies including the EuNetAir
project in Italy (Borrego et al., 2016) and EPA Community Air Sensor Network (CAIRSENSE) project (Jiao et al., 2016). Additionally, a handful of studies have tested the field performance of low-cost sensors both 'out of the box' with factory calibrations (Castell et al., 2017; Duvall et al., 2016), and after a machine-learning-based calibration (Cross et al., 2017; Esposito et al., 2016; Spinelle et al., 2015, 2017). The number of sensors and length of deployment used here is generally greater than those previous studies. We compare the performance of our RF models to these studies in Table 3. While several
low-cost sensor calibration studies have investigated calibration models within laboratory environments (Masson et al., 2015a; Mead et al., 2013; Piedrahita et al., 2014; Williams et al., 2013), we have elected to limit our comparison to field data.

There was not a substantial difference in performance of the RF model calibrated vs. LAB calibrated RAMP for CO, and performance was best for this pollutant on the 'out-of-the-box' factory calibrated performance assessments in EuNetAir and
CAIRSENSE, suggesting that rigorous calibration models may not be critical for CO. However, the RAMP CO RF model did provide improved performance (smallest MAE, 38 ppb) at lower average concentrations compared to the EuNetAir study. Similarly, the 'out-of-the-box' performance of the CO sensors tested as part of CAIRSENSE and by the 24 AQMesh sensors tested in Castell et al. (2017) was poorer than the RF model calibrated RAMP. Of those studies that used an advanced algorithm to calibrate the sensors (Cross et al., 2017; Spinelle et al., 2017), the CO RF model resulted in greater than or equivalent $R^2$
values and slightly lower slopes. While the $R^2$ of the CO HDMR model of Cross et al. (2017) is highest, it is difficult to estimate its true predictive performance due to its statistical metrics being calculated over the whole collocation period of which 35% of the data were used for training. Therefore, it blends goodness of fit and predictions.

For $NO_2$, the performance of 'out-of-the-box' low-cost sensors varied widely and half the sensors in the EuNetAir study
(Borrego et al., 2016) reported errors larger than the average ambient concentrations. Therefore, advanced calibration models, such as those using machine learning, are critical to accurate measurements of ambient $NO_2$. Furthermore, sensor performance was correlated with average ambient concentration; studies in areas with higher $NO_2$ concentrations had the best performance, consistent with our observations (Figure 10). For studies using advanced $NO_2$ sensor calibration models (Cross et al., 2017; Esposito et al., 2016; Spinelle et al., 2015), Esposito et al. (2016) had the best performance, with a MAE of < 2 ppb; however,
this evaluation was done in a location with high $NO_2$ concentrations, 45 ppbv (Air Quality England, 2015), more than three times higher than the 12 ppbv in Pittsburgh. In addition, they only evaluated one sensor array so the robustness of the approach is unknown. In our study, the MAEs across the $NO_2$ RF model RAMPs ranged from 2.6-3.8 ppb, which is almost as good as Esposito et al. (2016), but at less than one third the ambient concentrations. The slope and $R^2$ of the HDMR model for $NO_2$ of





Cross et al. (2017) do exceed that of the RAMP, but again their performance metrics appear to be calculated over the entire collocation period, which includes 35% training data. Similarly, the annual average $NO_2$ concentrations in 2015 were 15 ppb at the Massachusetts regulatory site used as a reference in Cross et al. (2017) (Massachusetts Department of Environmental Protection, 2016), 3 ppb higher than the average concentration observed in our study. As shown in Figure 10, an increase of a

few ppb of $NO_2$ can result in almost 100% reductions in relative residuals in our model, thus this effect is not surprising. Furthermore, for identical factory calibrated sensors out of the box, such as the Cairclip and AQMesh, a 5 ppb increase in average $NO_2$ concentration results in $R^2$ values more than doubling. As such, the excellent performance of the RF model for $NO_2$ at average ambient concentrations of 12 ppbv shows promise.

For $O_3$, the RF model, the calibrated data from Spinelle et al., (2015), and the measurements from the Aeroqual SM50 (Jiao et al., 2016) performed the best. Good performance from the Aeroqual when measuring $NO_2$ has also been previously observed (Delgado-Saborit, 2012). However, the results were the most consistent across the RAMP monitors calibrated with RF models, with relative standard deviations of <20% across the 19 RAMPs for all markers of statistical performance. This performance consistency also holds for the CO and $NO_2$ RF models. The $O_3$ RF models were built in Pittsburgh, PA, which has historically

had issues with NAAQS ozone compliance, thus while our model was seemingly one of the most accurate and robust, some of this performance may be attributed to the higher ambient $O_3$ concentrations. From this comparison, we conclude that the RAMP monitor calibrated with a RF model is unique in that it is more accurate when considering the combined suite of pollutants (i.e., all pollutants were accurately measured), it is consistent between many units (<20% relative standard deviation in performance metrics across 19 monitors), and is precise even at lower ambient concentrations.

**4.5 RF model calibrated RAMP performance in a monitoring context**

We further assess the RAMP monitor performance against two metrics: 1) for NAAQS compliance, and 2) for suitability for exposure measurements as per the US EPA Air Sensor Guidebook (Williams et al., 2014). We also demonstrate the benefit of improved performance of the RF models in a real-world deployment at two nearby sites in Pittsburgh, PA.

In this study, the time resolution and methods used to assess the effectiveness of the RF models (15 min) do not match the metrics used by regulators when considering compliance to National Ambient Air Quality Standards (NAAQS). For example, the NAAQS standard for $O_3$ is based on the maximum daily maximum 8-hour average, and compliance for $NO_2$ is based on the 98[th] percentile of the daily maximum 1-hour averages. While acknowledging that the RAMP monitor collocation period

was shorter than typical NAAQS compliance periods (e.g. annually for $O_3$ and across 3 years for $NO_2$) it is still worth characterizing the RAMP performance using the LAB, MLR and RF models (Figure 12). For the representative RAMP monitor used previously (RAMP #1), daily maximum 8-hour $O_3$ was in good agreement between the RF calibrated RAMP and the reference monitor, with all data points falling roughly along the 1:1 line, while for the MLR model, concentrations were



skewed slightly low (slope of 0.65 for MLR, 0.82 for RF). For NO$_2$, the 98$^{th}$ percentile of the daily maximum 1-hour averages was 34 ppb for the RF model versus 35 ppb measured using a reference monitor compared to 25 ppb for the MLR model and 51 ppb for the LAB model. The RF model was substantially closer to the reference monitor estimate and the underestimation was only by 1 ppb. Other RF model calibrated RAMP monitors performed similarly, all agreeing within 5 ppb.

To demonstrate the improved performance of the RF models in a real-world context, two of the RAMPs used in the evaluation study were deployed for a 6-week period at two nearby sites in Pittsburgh, PA. One RAMP monitor was located on the roof of a building at the Pittsburgh Zoo in a residential urban area, and another was placed approximately 1.5 km away at a near-road site located within 15 m of Highway 28 in Pittsburgh (Figure 13). NO$_2$ concentrations are known to be elevated up to 200

10    m away from a major roadway compared to urban backgrounds due to the reaction of fresh NO in vehicle exhaust with ambient O$_3$ (Zhou and Levy, 2007). Figure 13 shows the diurnal profiles of the RAMPs at the two locations evaluated using the RF and MLR models. The RF model indicates an NO$_2$ enhancement of approximately 6 ppb at the near-road site (Figure 13, red trace) compared to the nearby urban residential site (Figure 13, blue trace) and there are notable increases in NO$_2$ during morning and evening rush hour periods, as expected. The concentrations reported by the RF model calibrated RAMPs were further

verified with measurements using a mobile van equipped with reference instrumentation at different periods throughout the day. However, applying the MLR model to the RAMP data reveals no significant difference between the two sites (Figure 13, bottom diurnal). In fact, the MLR model predicts negative concentrations during the day. The results of this preliminary deployment suggest that the RF model calibrated RAMPs could be suitable for quantification of intra-urban pollutant gradients.

The US EPA Air Sensor Guidebook (Williams et al., 2014) provides air sensor performance goals by application area . The performance criteria include maximum precision and bias error rates for applications ranging from education and information (Tier I) to regulatory monitoring (Tier V). The precision estimator is the upper bound of a 90% confidence interval of the coefficient of variation (CV) and the bias estimator is the upper bound of a 95% confidence interval of the mean absolute percent difference between the sensors and the reference (full equations in the Supplemental Information). An overarching

goal of RAMP monitor deployments is to use low-cost sensor networks to quantify intra-urban exposure gradients, thus our benchmark performance was Tier IV (Personal Exposure), which recommends that low-cost sensors have precision and bias error rates of less than 30%. For the testing (withheld) periods, we compared the performance of the RF, MLR and LAB models for all the RAMP monitors used in this study to the precision and bias estimators recommended by the US EPA (Figure 14). The performance across the RAMP monitors was summarized using box plots, and only the RF model calibrated RAMPs

are suitably precise and accurate for Tier IV (personal exposure) monitoring across CO, NO$_2$ and O$_3$. Furthermore, both RF model calibrated CO and O$_3$ RAMP monitor measurements were below the even more stringent Tier III (Supplemental Monitoring) standards, which recommends precision and bias error rates of <20%. The RF model NO$_2$ RAMP measurements may reach Tier III in locations with higher NO$_2$ concentrations.





## 5 Conclusions

This study demonstrates that the RF model applied to the RAMP low-cost sensor package can accurately characterize air pollution concentrations at the low levels typical of many urban areas in the United States and Europe. The fractional error of the models at a 15-minute time resolution was <5% for $CO_2$, approximately 10-15% for CO and $O_3$ and approximately 30%

for $NO_2$, corresponding to mean absolute errors of 10 ppm, 38 ppb, 3.4 ppb and 3.5 ppb, respectively. This performance meets the recommended precision and accuracy error metrics from the US EPA Air Sensor Guidebook for Personal Exposure (Tier IV) monitoring. We demonstrate that degree of sensitivity allows quantification of intra-urban gradients. Furthermore, the calibration models were well-constrained across 19 RAMP units (all performance metrics <20% relative standard deviation), and showed minimal degradation over the duration of the collocation study (August 2016 – February 2017),

While the iteration of the RAMP used in this study was equipped with an $SO_2$ sensor, no calibration model was possible due to $SO_2$ concentrations at our supersite being below reference instrument detection limits. One feature of the RAMP monitor is that the sensors are modular and can be readily replaced. The assessment of explanatory variable importance combined with the sub-detection limit levels $SO_2$ during the study suggests that the RAMP monitor did not benefit from the presence of the

$SO_2$ sensor in this urban background environment. Future iterations of the RAMP will be equipped with NO sensors, which may be more relevant in an urban context.

The RF-models described here were built on four weeks of training data equally distributed in 3.5 day periods throughout the entire collocation. This is nominally equivalent to 3-4 days of calibration every 2 months. As previously mentioned, the low-

cost sensor modules within the RAMP monitors can be readily replaced, and as such, we recommend for a large urban deployment to prepare a set of sensors at a regulatory monitoring site and to exchange sensors as they malfunction or as calibration models drift. Since the completion of this study, the sensors have been deployed in Pittsburgh for over 4 months, and changes in the calibration models over longer periods of deployment (1 year or more) will be discussed in a future work. Additionally, the sensors were first opened in July 2016, and characterized over the first 7 months of exposure to ambient

environments. During this period, no significant temporal drift or sensor degradation was observed, but longer observational studies are likely needed to characterize sensor decay and end-of-life.

The calibration models were developed in Pittsburgh, which had higher $O_3$ and lower $NO_2$ compared to several published field-based calibrations and measurements with low-cost sensors. Our results and those of other studies demonstrate that low-cost

sensor performance generally increases with increasing ambient concentration, but despite this, the RF models for $NO_2$ had the second lowest mean absolute error (<4 ppbv) even at low $NO_2$ concentrations. The good performance of the RF models across all pollutants can likely be attributed to the ability of the RF models to account for pollutant and meteorological cross-sensitivities, highlighting the importance of building multipollutant sensor packages.





Overall, we conclude that with careful data management and calibration using advanced machine learning models, that low-cost sensing with the RAMP monitors may significantly improve our ability to resolve spatial heterogeneity in air pollutant concentrations. Developing highly resolved air pollutant maps will assist researchers, policymakers and communities in developing new policies or mitigation strategies to enhance human health. Going forward, a random forest calibrated RAMP network of up to 50 nodes will be deployed in Pittsburgh, PA. This robustly calibrated network will help support better epidemiological models, aid in policy planning, and identify areas where more assessment is needed.

**Competing interests**

Author J. Gu is the CEO of SenSevere, the developer and manufacturer of the RAMP hardware. The extent of J. Gu's involvement was solely in development management, and improvement of the hardware in the RAMP monitors, and not in data analysis. Authors N. Zimmerman and R. Subramanian may in the future act as consultants for SenSevere on low-cost sensor calibration. The data output from the SenSevere hardware in conjunction with the calibration algorithms presented in this paper yields significantly more accurate measurements than previously reported, and are the subject of provisional patent application. The authors declare no other competing interests.

**Acknowledgements**

 Funding for this study was provided by the Environmental Protection Agency (Assistance Agreement Nos. RD83587301 and 83628601), and the Heinz Endowment Fund (Grants E2375 and E3145). N. Zimmerman's funding was provided by the NSERC Postdoctoral Fellowship (PDF-487660-2016). The authors also wish to thank A. Ellis and J.S. Apte for helpful conversations, and M. Schurman Boehm for her assistance with the laboratory calibrations.

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

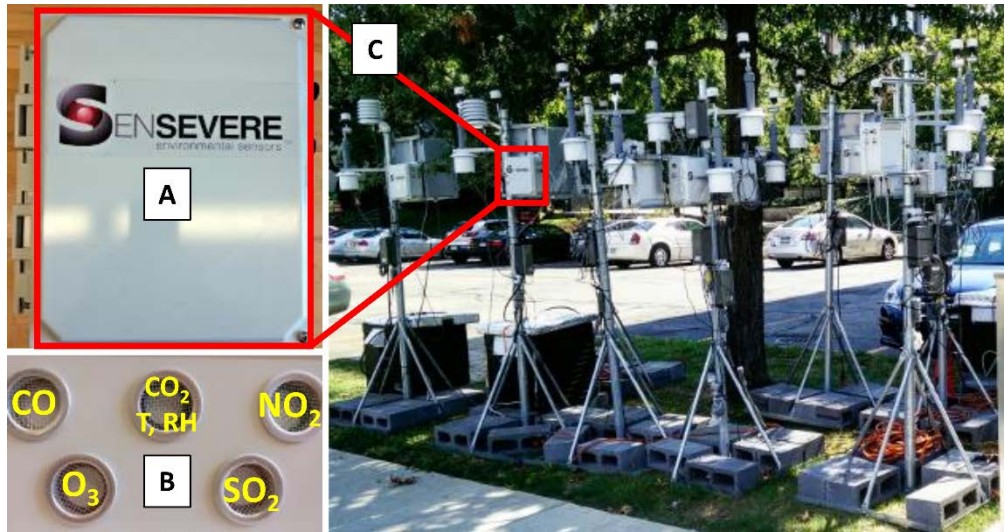

**Figure 1: Photographs of the RAMP monitors and the sampling set up. (A) Front view of the RAMP unit in the NEMA-rated enclosure. (B) Bottom view of the RAMPs with sensor layout labelled in yellow. (C) Example of collocation set-up using tripod mounting (not pictured: supersite containing the reference monitors, immediately beside the tripods).**

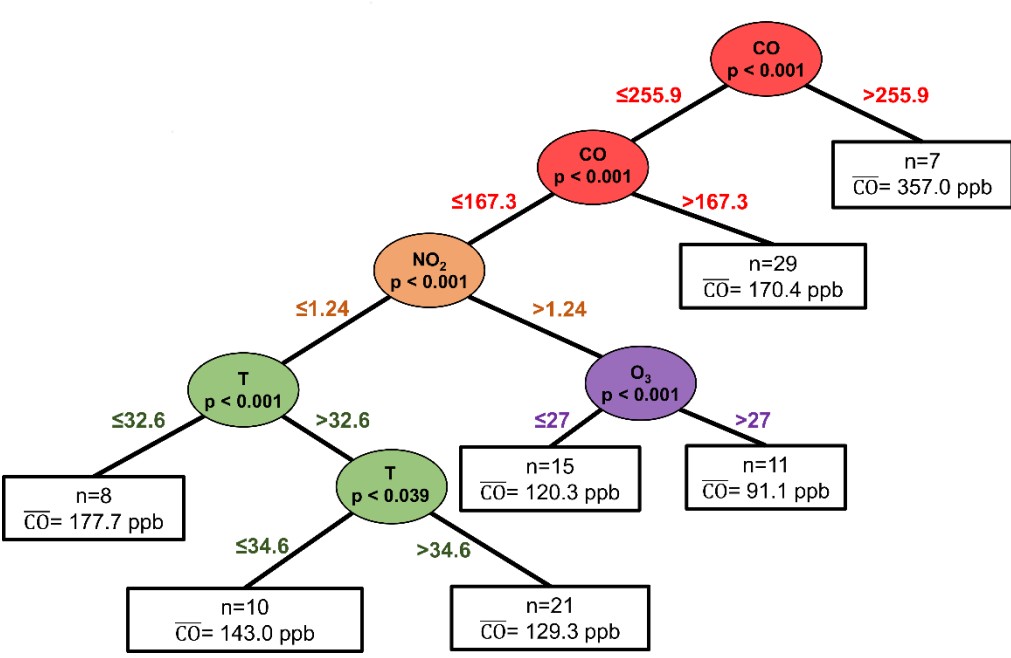

**Figure 2: Simplified illustration of one potential CO random forest tree for one RAMP using 100 data points (the trees within the actual models are significantly more complex and 500 such trees are included in the final models). Tree nodes are coloured by splitting variable and split point is overlaid on the branch (e.g., at first split, points with CO sensor signal >255.9 are sent to a terminal node, the remaining points go to the next splitting node). $\overline{CO}$ is the average CO reference monitor concentration (ppb) in each terminal node; n = number of data points in each terminal node.**





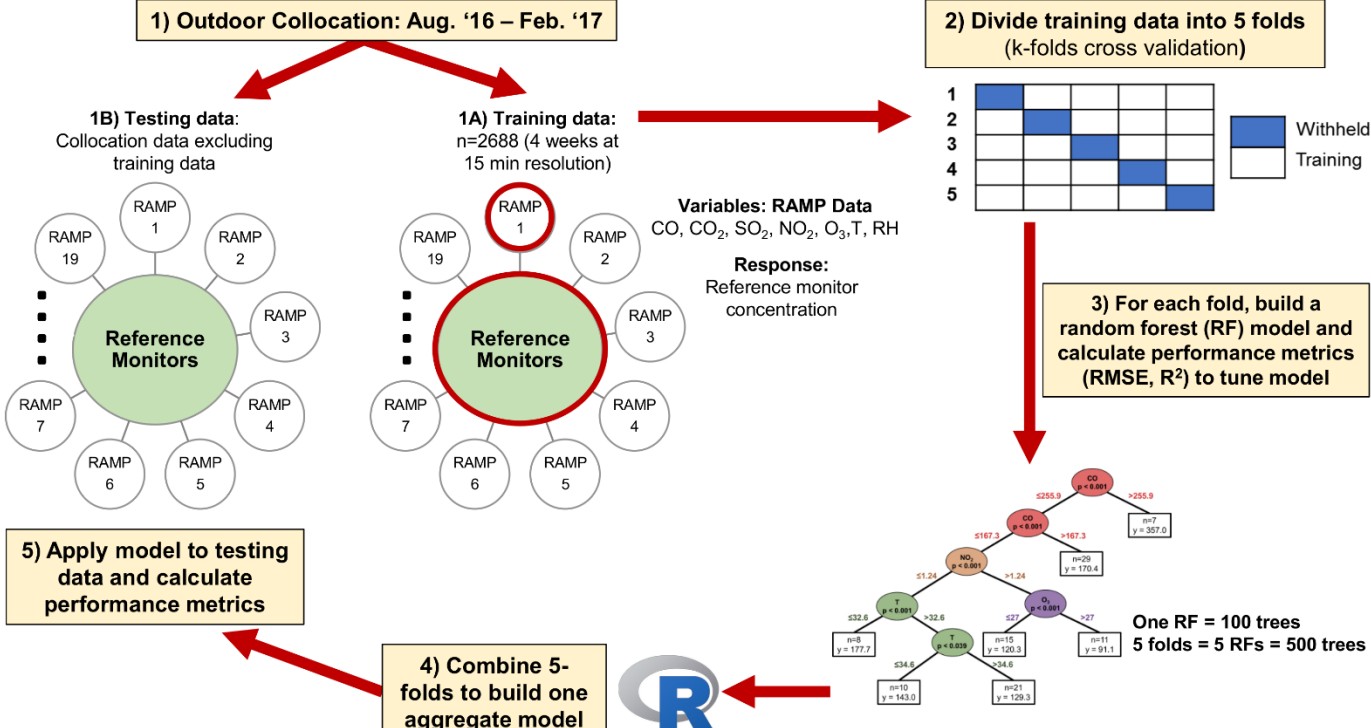

**Figure 3:** Flow path for data collection and RF model fitting and testing. From collocation period, 2688 points were sub-selected as training (1A) data while the remaining data were used for model testing (1B). The training data were further divided into 5 cross-validation folds and each fold was used to tune and build an RF model. All five models were then combined in R to build one cumulative model and the predictive power of the model was assessed for the withheld testing data.





**Figure 4: Example time series and regressions comparing the reference monitor data (black) to statistically average RAMP (RAMP#1) using LAB model (green), multiple linear regression (MLR) model (blue) and random forest (RF) model (pink). This example shows only 48 hrs of time series data to illustrate approach; the full evaluations (Table 3) were performed with much larger testing datasets.**





**Figure 5: RF model performance for ozone evaluated using the testing data (data withheld from building model). Correlation plots show predicted ozone concentration ("RAMP") versus the reference monitor concentration ("REF") for 16 RAMP units. All values are in ppb, and the 1:1 line is drawn as a black dashed line.**




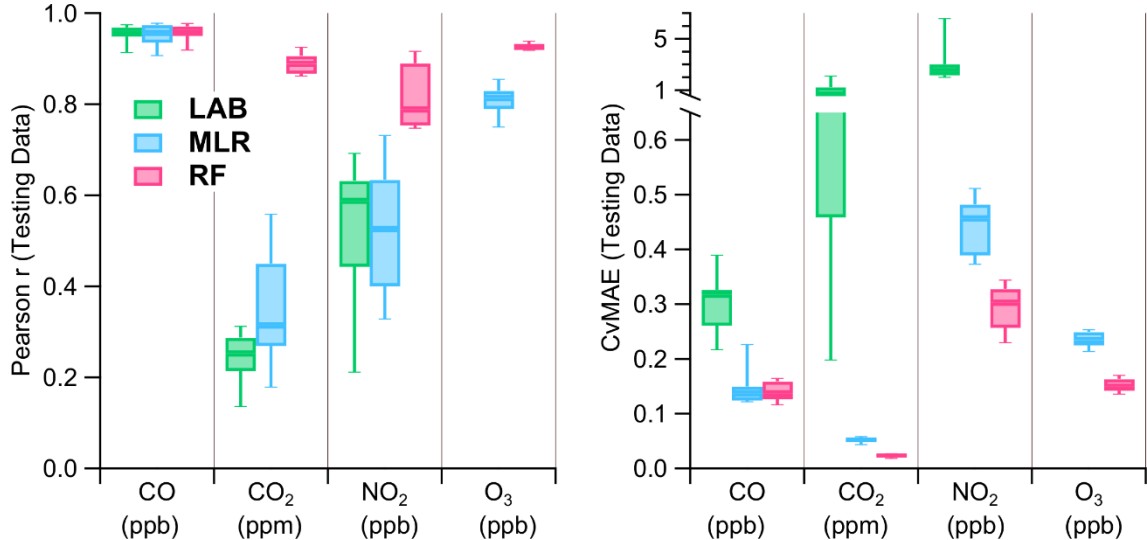

**Figure 6: Performance of different calibration models against reference monitor testing data (data not included in model fitting).** *Left:* **Pearson r correlation coefficient (higher = better, maximum of 1) of different calibration models '(LAB', green; 'MLR', blue; 'RF', pink) versus reference monitor.** *Right:* **The CvMAE (coefficient of variation of the MAE; MAE normalized by average reference concentration, lower = better) for the three calibration methods. The box plots show the range across the 19 RAMPs (whiskers: 10th and 90th percentile, box edges: 25th and 75th percentile).**





**Figure 7: Target diagrams for CO, CO₂, NO₂ and O₃ to compare the LAB, MLR and RF model performance. The y-axis is the bias relative to the reference and the x-axis is the bias-adjusted RMSE (CRMSE) normalized by reference monitor standard deviation; the vector distance between any given point and the origin is the RMSE normalized by the standard deviation of the reference measurements. The CRMSE is in the left plane if model standard deviation is smaller than the standard deviation of the reference observations, and vice versa. If data falls within the circle, then the variance of the residuals is smaller than the variance of the reference measurements. The target diagram for the LAB model for CO₂ and NO₂ is shown in the inset figure because of the order of magnitude difference in MBE and CRMSE compared to the MLR and RF models.**





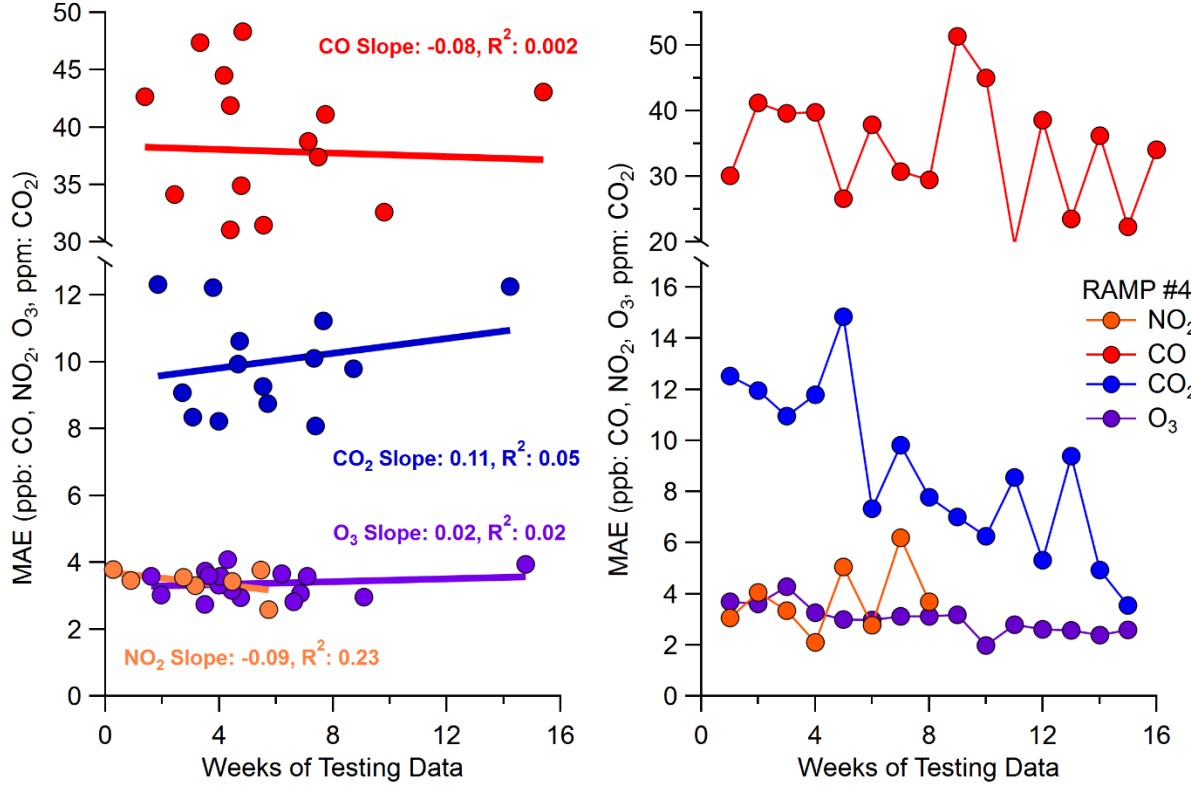

**Figure 8:** *Left:* **Mean absolute error (MAE) versus the length of the testing period for CO (red), CO₂ (blue), NO₂ (orange) and O₃ (purple) for all the RAMPs.** *Right:* **Changes in MAE over time for the RAMP with the longest testing window (RAMP #4). The figure shows that the MAE is generally unchanged (or in some cases improves) as the amount of testing data increases, suggesting the RF models are stable over long periods.**



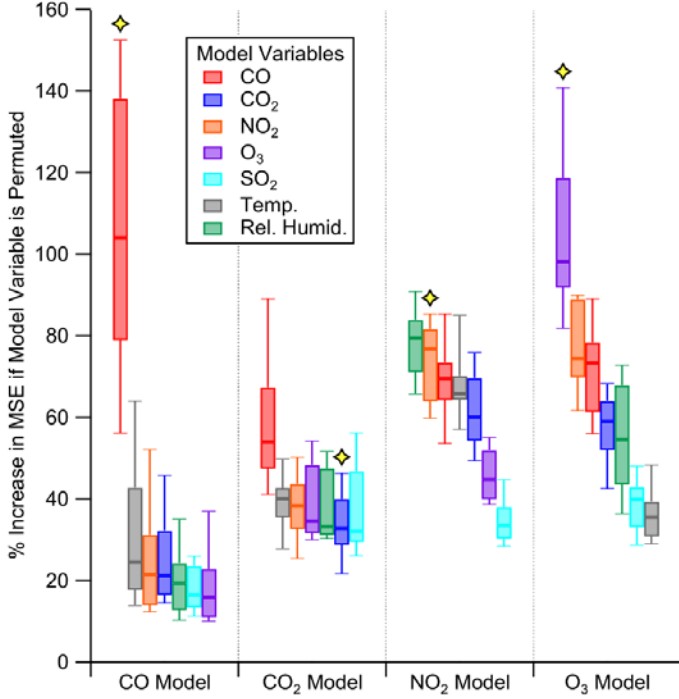

**Figure 9: Importance of the explanatory variables to each of the RF models. For each model, the explanatory variables are rank ordered from most to least important, and the sensor response corresponding to the target analyte is marked with a yellow star. The box plots represent the range of importance across the 19 RAMPs (whiskers: 10th and 90th percentile, box edges: 25th and 75th percentile). The relative importance is determined by calculating the increase in mean square error if the explanatory variable is permuted (i.e., randomly shuffled).**





**Figure 10: Box plots from the 19 RAMP monitors of median concentrations measured by monitors (bottom) and median model residuals (middle) and model residuals normalized to the reference concentration (top) for each pollutant, divided into deciles. The box plots provide the range of medians by the different RAMP monitors.**





**Figure 11: Illustrating the range of predictions from the 500 trees for RAMP #1. The testing data were binned and averaged. The concentration measured by the reference monitors is then plotted against the average concentration from the model. The error bars represent the standard deviation of the answers from the 500 trees and the bins are colour coded by the number of data points within each bin. The dashed black line is the 1:1 line.**




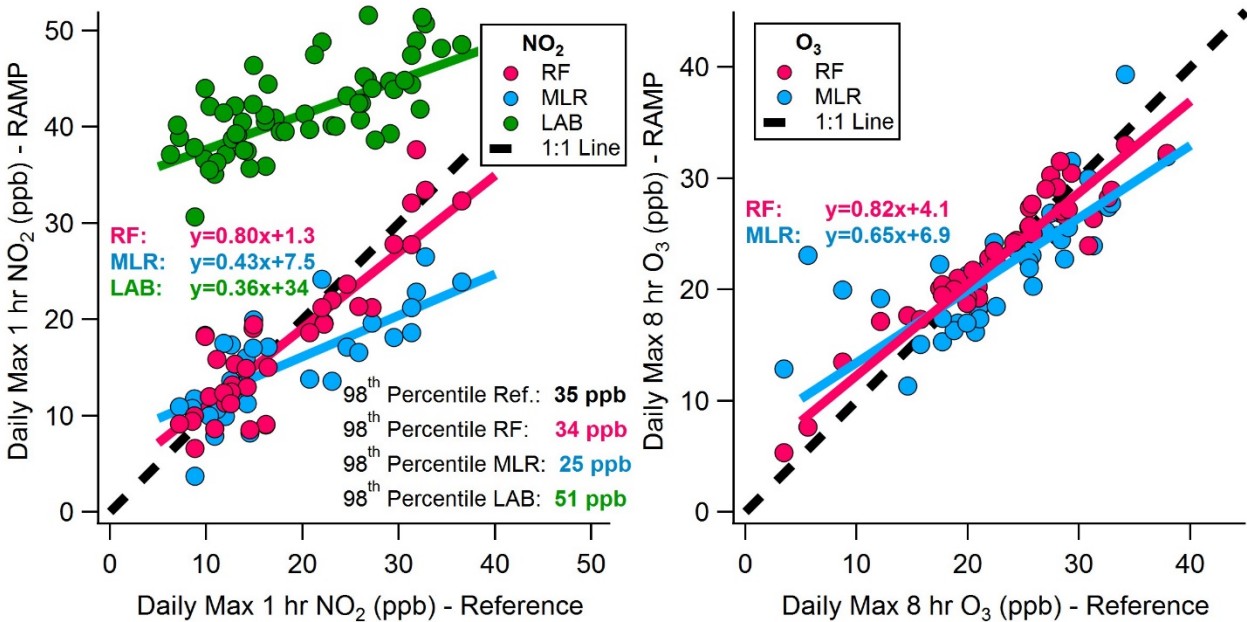

**Figure 12: Performance of one representative RAMP (RAMP#1) for NAAQS compliance metrics (O₃: Daily Max 8 h, NO₂: 98th percentile of Daily Max 1 h averages) Right: comparison of daily 8 hr maximum reference monitor ozone concentrations (x-axis) to MLR and RF models. Left: comparison of daily 1 h maximum reference monitor concentrations versus the LAB, MLR and RF models. The NO₂ standard is the 98th percentile of the daily 1 h maximums.**

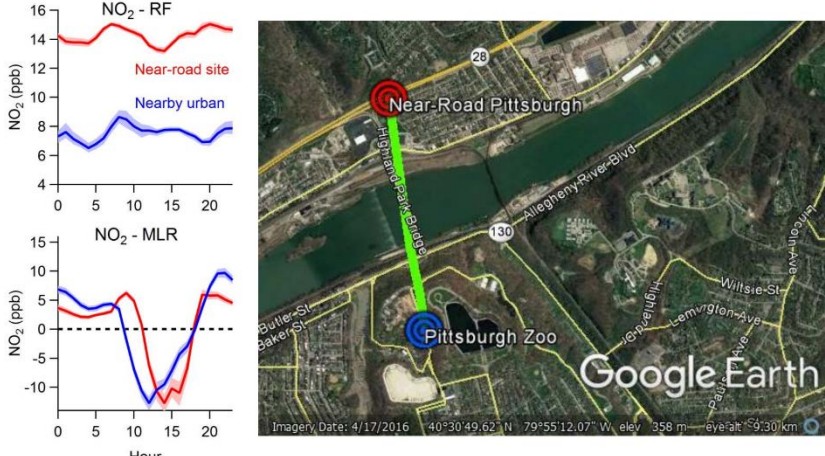

**Figure 13: Left: Diurnal NO₂ patterns at two nearby sites (one urban, one near-road) measured by RAMP monitors calibrated using RF models (top) or MLR models (bottom), Right: Satellite view of the two sites, which were ~1.5 km apart. The urban site was at the Pittsburgh Zoo and the near-road site was within 15 m of Highway 28.**



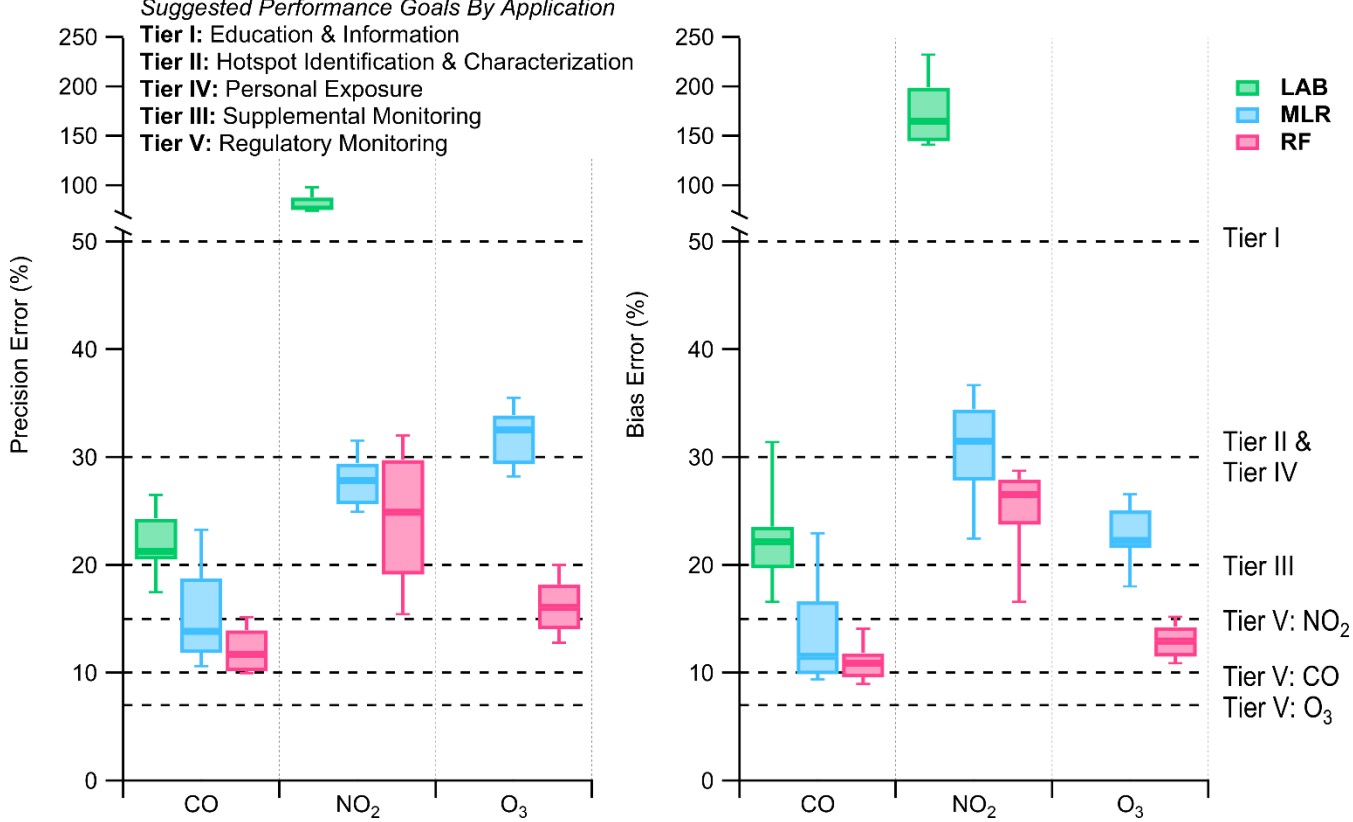

**Figure 14: Precision (left) and bias (right) estimates of RAMP monitors calibrated using LAB, MLR, and RF models compared to the suggested performance goals by application as recommended in the EPA Air Sensor Guidebook. The precision estimator is the upper bound of the coefficient of variation (upper bound of the relative standard deviation, RSD). The box plots are the range of performance across the calibrated RAMP monitors (testing data only). The calibrated RAMP monitors meet the recommended error limits for exposure (Tier IV).**





**Table 1: Calibration ranges for laboratory-based calibration (LAB)**

| Pollutant | Calibration Range | Points per Calibration |
|---|---|---|
| CO | 0 – 1600 ppb | 3-4 |
| $NO_2$ | 0 – 50 ppb | 3-4 |
| $CO_2$ | 0 – 500 ppm | 3-4 |

**Table 2: Performance metrics for fits to training data.**

| Type | Species | # RAMPs | Avg. Pearson r (±SD) | Avg. MAE (±SD) | Avg. CvMAE (±SD) | $\beta_0$ (±SD) | $\beta_1$ (±SD) | $\beta_2$ (±SD) | $\beta_3$ (±SD) |
|---|---|---|---|---|---|---|---|---|---|
| LAB | CO | 9 | 0.99 (±0.01) | 132 (±32 ppb) | 38% (±17%) | -119 (±53) | 0.82 (±0.69) | - | - |
| | $CO_2$ | 14 | 0.99 (±0.01) | 28 (±24 ppm) | 24% (±12%) | 20 (±36) | 0.98 (±0.13) | - | - |
| | $NO_2$ | 14 | 0.99 (±0.01) | 35 (±8 ppb) | 188% (±48%) | -14 (±4.9) | 0.62 (±0.15) | - | - |
| **Type** | **Species** | **# RAMPs** | **Avg. Pearson r (±SD)** | **Avg. MAE (±SD)** | **Avg. CvMAE (±SD)** | **$\beta_0$ (±SD)** | **$\beta_1$ (±SD)** | **$\beta_2$ (±SD)** | **$\beta_3$ (±SD)** |
| MLR | CO | 19 | 0.94 (±0.06) | 39 (±13 ppb) | 15% (±5%) | 32 (±50) | 1.3 (±0.2) | -1.1 (±2.8) | -0.1 (±0.6) |
| | $NO_2$ | 19 | 0.59 (±0.17) | 4.6 (±0.7 ppb) | 42% (±5%) | 3.9 (±16) | 1.2 (±0.5) | 0.1 (±0.3) | -0.1 (±0.2) |
| | $O_3$ | 19 | 0.81 (±0.06) | 5.1 (±0.6 ppb) | 24% (±2%) | 9.4 (±14) | 0.92 (±0.2) | 0.1 (±0.2) | -0.2 (±0.2) |
| | $CO_2$ | 19 | 0.49 (±0.13) | 19 (±3 ppm) | 4% (±1%) | 390 (±72) | 0.1 (±0.1) | -0.8 (±0.7) | 0.1 (±1.0) |
| **Type** | **Species** | **# RAMPs** | **Avg. Pearson r (±SD)** | **Avg. MAE (±SD)** | **Avg. CvMAE (±SD)** | **Median $m_{try}$** | **$m_{try} = 2$** | **$m_{try} = 4$** | **$m_{try} = 7$** |
| RF | CO | 19 | 0.99 (±0.00) | 7.9 (±1.5 ppb) | 3% (±0.5%) | 7 | 11% | 21% | 68% |
| | $NO_2$ | 19 | 0.99 (±0.01) | 0.5 (±0.1 ppb) | 5% (±1%) | 4 | 21% | 74% | 5% |
| | $O_3$ | 19 | 0.99 (±0.00) | 0.7 (±0.1 ppb) | 3% (±0.4%) | 4 | 0% | 84% | 16% |
| | $CO_2$ | 19 | 0.99 (±0.00) | 1.7 (±0.3 ppm) | 0.4% (±0.1%) | 2 | 74% | 21% | 5% |

LAB: Laboratory calibration (Eq. 1), MLR: multiple linear regression (Eq. 2), RF: random forest model.
For the LAB and MLR models, the fit coefficients are provided.
For the RF models, the median mtry value across the 19 RAMPs and the breakdown of the mtry tuning results ($m_{try}$ which minimized RMSE) across the 19 RAMPs results are provided.





**Table 3: Comparison to other published studies.**

| | Project | Location | Sensor Node | Type | N (days) | AvgConc (ppb) | Slope | R² | MAE (ppb) | MBE (ppb) |
|---|---|---|---|---|---|---|---|---|---|---|
| **CO** | EuNetAir[1] | Aveiro, PT | AirSensorBox | EC | 6 | 330 | NR | 0.76 | 90 | 0 |
| | EuNetAir[1] | Aveiro, PT | NanoEnvi | EC | 9 | 330 | NR | 0.53 | 100 | 100 |
| | EuNetAir[1] | Aveiro, PT | Cambridge CAM11 | EC | 14 | 330 | NR | 0.87 | 180 | -200 |
| | EuNetAir[1] | Aveiro, PT | AQMesh | EC | 15 | 330 | NR | 0.86 | 50 | 0 |
| | CAIRSENSE[2] | Decatur, GA, US | AQMesh | EC | 110-111 | 330 | NR | 0.77-0.87 | NR | NR |
| | CAIRSENSE[2] | Decatur, GA, US | Air Quality Egg | MOS | 115-196 | 310 | NR | <0.25 | NR | NR |
| | Castell et al.[3] | Kirkeveien, NO | AQMesh | EC | 72 | NR | 0.88* | 0.36 | 150 | -150 |
| | Spinelle et al.[4] | Ispra, IT | Figaro, e2V | EC, MOS | 85 | 230 | 1.01-1.38 | 0.29-0.37 | NR | NR |
| | Cross et al.[5] | Boston, MA, US | ARISense | EC | ~120** | -- | 0.96** | 0.96** | NR | NR |
| | **This Study** | **Pittsburgh, PA, US** | **RAMP** | **EC** | **41 [10-108]** | **270 (±30)** | **0.86 (±0.09)** | **0.91 (±0.05)** | **38 (±6.5)** | **0.1 (±0.2)** |
| **NO₂** | EuNetAir[1] | Aveiro, PT | Cambridge CAM11 | EC | 14 | 16 | NR | 0.84 | 5.61 | -2.3 |
| | EuNetAir[1] | Aveiro, PT | AirSensorBox | EC | 7 | 16 | NR | 0.06 | 20.2 | 17.7 |
| | EuNetAir[1] | Aveiro, PT | NanoEnvi | EC | 7 | 16 | NR | 0.57 | 14.9 | 13.1 |
| | EuNetAir[1] | Aveiro, PT | ECN_Box_10 | EC | 11 | 16 | NR | 0.89 | 4.95 | -1 |
| | EuNetAir[1] | Aveiro, PT | AQMesh | EC | 6 | 16 | NR | 0.89 | 1.46 | 0 |
| | EuNetAir[1] | Aveiro, PT | ISAG | MOS | 13 | 16 | NR | 0.02 | 16.2 | 349.5 |
| | CAIRSENSE[2] | Decatur, GA, US | Cairclip | EC | 194-285 | 11 | 0.96 | <0.25-0.57 | NR | NR |
| | CAIRSENSE[2] | Decatur, GA, US | AQMesh | EC | 110-111 | 10 | NR | <0.25 | NR | NR |
| | CAIRSENSE[2] | Decatur, GA, US | Air Quality Egg | MOS | 115-196 | 11 | NR | <0.25 | NR | NR |
| | Duvall et al.[6] | Houston, TX, US | Cairclip | EC | 24 | 5.5 | 0.25 | 0.01 | NR | NR |
| | Duvall et al.[6] | Denver, CO, US | Cairclip | EC | 30 | 5.1 | 0.04 | <0.01 | NR | NR |
| | Castell et al.[3] | Kirkeveien, NO | AQMesh | EC | 72 | NR | 0.2-0.38* | 0.24 | 26.2 | 13.3 |
| | Esposito et al.[7] | Cambridge, UK | SnaQ | EC | 28 | NR | NR | 0.83 | 1.27 | NR |
| | Spinelle et al.[8] | Ispra, IT | αSense, Citytech | EC | 86 | 9 | 0.64-0.79 | 0.55-0.59 | NR | NR |
| | Cross et al.[5] | Boston, MA, US | ARISense | EC | ~120** | NR | 0.83** | 0.80** | NR | NR |
| | **This Study** | **Pittsburgh, PA, US** | **RAMP** | **EC** | **24 [2-56]** | **12 (±1.4)** | **0.64 (±0.11)** | **0.67 (±0.12)** | **3.48 (±0.36)** | **-0.4 (±1.13)** |
| **O₃** | EuNetAir[1] | Aveiro, PT | AirSensorBox | EC | 6 | 17 | NR | 0.13 | 22.12 | 19.2 |
| | EuNetAir[1] | Aveiro, PT | NanoEnvi | MOS | 9 | 17 | NR | 0.77 | 7.66 | 6.5 |
| | EuNetAir[1] | Aveiro, PT | Cambridge CAM11 | EC | 11 | 17 | NR | 0.14 | 21.5 | 15.7 |
| | EuNetAir[1] | Aveiro, PT | AQMesh | EC | 6 | 17 | NR | 0.7 | 2.4 | 0 |
| | EuNetAir[1] | Aveiro, PT | ISAG | MOS | 13 | 17 | NR | 0.12 | 360.12 | 356.1 |
| | CAIRSENSE[2] | Decatur, GA, US | Aeroqual SM50 | GSS | 168-281 | 18 | 0.81-0.96 | 0.82-0.94 | NR | NR |
| | CAIRSENSE[2] | Decatur, GA, US | Cairclip | EC | 194-285 | 17 | 0.68-0.85 | 0.68-0.88 | NR | NR |
| | CAIRSENSE[2] | Decatur, GA, US | AQMesh | EC | 110-111 | 15 | NR | <0.25 | NR | NR |
| | Duvall et al.[6] | Houston, TX, US | Cairclip | EC | 24 | 32 | 0.93 | 0.80 | NR | NR |
| | Duvall et al.[6] | Denver, CO, US | Cairclip | EC | 30 | 46 | 1.19 | 0.77 | NR | NR |
| | Castell et al.[3] | Kirkevein, NO | AQMesh | EC | 72 | NR | 0.11-0.26* | 0.29 | 19.9 | 6.8 |
| | Esposito et al.[7] | Cambridge, UK | SnaQ | EC | 28 | NR | NR | 0.69 | 7.45 | -- |
| | Spinelle et al.[8] | Ispara, IT | αSense, Citytech | EC | 82-84 | 30 | 1.02-1.12 | 0.86-0.91 | NR | NR |
| | Cross et al.[5] | Boston, MA, US | ARISense | EC | ~120** | NR | 0.62** | 0.51** | NR | NR |
| | **This Study** | **Pittsburgh, PA, US** | **RAMP** | **EC** | **38 [11-103]** | **22 (±1.4)** | **0.82 (±0.05)** | **0.86 (±0.02)** | **3.36 (±0.41)** | **-0.14 (±0.46)** |

[1](Borrego et al., 2016), [2](Jiao et al., 2016), [3](Castell et al., 2017), [4](Spinelle et al., 2017), [5](Cross et al., 2017), [6](Duvall et al., 2016), [7](Esposito et al., 2016), [8](Spinelle et al., 2015)

EC=electrochemical, MOS=metal oxide sensor, GSS=gas sensitive semiconductor. NR= not reported in manuscript.

For RAMP data, bracketed data is range (for N days) or standard deviation (all other metrics) across all the RAMP units.

*values for slopes only provided for a subset of 2 of 24 sensors

**Cross et al. performance metrics reported for full collocation dataset which includes both testing (65% of data) and training (35%) of data. Performance metrics for other studies are only based on testing data not used for model fitting/training.