# Peer review of "A machine learning calibration model using random forests to improve sensor performance for lower-cost air quality monitoring"

_Atmospheric Measurement Techniques, 2017_

## Referee Comment (RC1) · Anonymous Referee #1 · 11 Sep 2017

This work presents calibration models for low-cost sensors of NO2, O3, CO, and CO2 that show encouraging results toward use of such devices in exposure analysis and dense measurement networks. The authors describe the instruments, calibration algorithm, and evaluation metrics in a pedagogical way that is easy to follow. The most advanced algorithm (Random Forest) performed the best, and from it the authors were able to extract the importance of each variable to the calibration to provide better understanding of these statistical models. The authors place the performance of their new calibrations in context of simpler calibration models, previous studies on sensor calibration, and several performance guidelines established by regulatory agencies. The machine learning approach appears to take into account interfering cross-sensitivities

to other pollutants, but at times predict by correlation (e.g., for CO2 and NO2) rather than direct signal response to the pollutant. The manuscript is well-written and explains an approach that allows identification of important explanatory variables for a statistical calibration model, shows that several calibration models are able to maintain reasonable performance even in lower concentrations than demonstrated in previous studies. The manuscript presents a relevant contribution to the emerging literature on low-cost sensor calibration and is therefore recommended for publication in Atmospheric Measurement Techniques after the following comments have been addressed.

General comments:

The title is a bit ambitious, ambiguous, or both. How much of the performance "gap" is closed by a) improved hardware compared to past studies, b) the algorithm (i.e., Random Forest), c) sensor combinations at each node, and d) range of different sample types collected? Application of machine learning for sensor calibration in the field has been performed before, but the title and abstract seems to give the impression that this reduces the gap. There is much focus given to RF but there is no indication that it has an inherent advantage over other machine learning methods. For instance, it is possible that a MLR model could also handle cross-sensitivities only if it were provided all variables (though RF and other machine learning algorithms are more flexible in that it does not require the assumption regarding global linearity).

The past work of De Vito et al. (2008, 2009) also show encouraging results from a long-term evaluation of field calibrations (for low-cost multi-sensor devices for benzene, CO, and NO2 against government monitoring station instruments using machine learning algorithms).:

De Vito S., Massera E., Piga M., Martinotto L., and Di Francia G.: On field calibration of an electronic nose for benzene estimation in an urban pollution monitoring scenario, Sensors and Actuators B: Chemical, 129(2):750–757, doi:10.1016/j.snb.2007.09.060, 2008.

De Vito S., Piga M., Martinotto L., and Di Francia G.: CO, NO2 and NOx urban pollution monitoring with on-field calibrated electronic nose by automatic bayesian regularization, Sensors and Actuators B: Chemical, 143(1):182–191, doi:10.1016/j.snb.2009.08.041, 2009.

The manuscript is perhaps too bold in its tone. Accurate predictions are shown for concentration (and T, RH) domains that are present at the location of the reference monitor used for calibration, even while using different data points. (As stated by the authors, current implementation of RF is limited to the domain of the training set.) Dense network coverage implies monitor placement in different microenvironments (e.g., near-roadway, etc.) which would experience different concentration regimes. Moreover, some of the explanatory variables used for calibration may be surrogates for another variable which may vary differently at another site. There is mention of two RAMPS units deployed in Pittsburgh and their positive evaluation against other reference measurements in a mobile van (p. 17, line 15), but no results are shown.

Since corrections of the supersite reference monitors against the Allegheny County Health Department instruments are necessary, why not make this Allegheny County Health Department site the reference site? Given the local contributions of vehicle emissions to CO and NO2 that are present in the parking lot site, how were the corrections for baseline drift determined?

While the authors describe the use of 5-fold CV to selection the explanatory variables to use, the choice of 5 data points per terminal node / 100 trees per fold does not seem to be explained. This was also selected in the CV process?

p. 14 Line 18 paragraph: Is this not possibly a limitation of the hardware?

Minor comments:

Section 2.2: Data coverage (i.e., missing data) and the time resolution should be stated here rather than (or in addition to) later in the manuscript.

[Figure]

P. 9 Line 15 to end of paragraph. The authors switch from describing "intermittent" collocation to "distributed" collocation. Given the discussion of multiple RAMP monitors, "distributed" can be confusing. Also, "degree of collocation" is referring to frequency or effective duration?

p. 10 Line 19: value of correlation for NO2 and CO2 with reference monitors is missing.

p. 10 Line 22: insert figure numbers (SI Fig S3-S6).

p. 10 Line 30: The relationship between m_try and model complexity is not very clear.

p. 10 Line 13: "clearly outperformed" -> not for CO

p. 11 Line 21: insert figure numbers (SI Figs S7-S10). Slopes, correlations, or some of the metrics listed in Table S2 included in the panels would be informative. Why are some RAMPS not included?

p. 11 Line 31: "NO2" -> "O3" here?

---

## Referee Comment (RC2) · Anonymous Referee #2 · 17 Sep 2017

This paper explores the performance of a low-cost sensor unit for measurement of urban air quality. A total of nineteen multi-pollutant sensor packages (called RAMP) which measure CO, $NO_2$, $O_3$ and $CO_2$ as well as rH and Temp have been used from August 2016 through February 2017 in Pittsburgh PA next to an air quality monitoring site where reference instruments have been operated. Measurements from the reference instruments have been used as independent variables for investigation of different models for the calibration of the low-cost sensor units. The responses from all sensors in the RAMP units have been used for prediction of the pollutant concentration. It was found that calibration models based on a machine learning technique (Random Forests, RF) performed much better than (multiple) linear regression mod-

els. The authors find that the combination of the RF calibration approach and the multi-pollutant sensor package accounts for pollutant cross sensitivities and is a promising approach for the use of low-cost air quality sensors. The manuscript covers a relevant and emerging topic and adds to a growing number of studies on sensor calibration and performance of low-cost sensors. However, there are some technical flaws in the manuscript that should be corrected. The manuscript can be published in AMT after consideration of the following comments.

The overall message of the manuscript is in my view too optimistic and can for readers be misleading. The authors should make clear that the good performance of the sensors found in this calibration study does not imply that the sensor unit is capable of providing similarly accurate air quality measurements in a real-world application. A good performance of sensor units in a calibration exercise like the study at hand is certainly necessary but not sufficient for the suitability of the sensors for real world air quality measurements. It should be clear that the manuscript is targeting on the good data quality obtained when combining the multi-pollutant sensor unit and RF and that a full assessment of the performance of the RAMPs within a sensor network for air quality measurements under real world conditions requires future research (and solutions for the quality assurance and quality control of the deployed sensors). The authors touch this point briefly in the conclusions section, however, for readers the impression remains that the RAMPS sensor units are ready for being used for urban air quality assessments. For example, in the conclusions section, last paragraph, it is stated that "Overall, we conclude that with careful data management and calibration using advanced machine learning models, that low-cost sensing with the RAMP monitors may significantly improve our ability to resolve spatial heterogeneity in air pollutant concentrations.". This conclusion is not justified by the available study and should be kept for the future work when results on the data quality as obtained in real world applications are available. As another example, the authors write on page 14, lines 14-16 "The US EPA limit of detection for federal regulatory monitors is 10 ppb for both NO2 and O3, suggesting that as with CO, the RF model performance is within 20% of

regulatory standards (United States Environmental Protection Agency, 2014)". This is again misleading: It can be concluded from this calibration study that the performance of sensors with an updated calibration meet those requirements, the data quality that can be achieved with the sensor under real world conditions is something different and currently not known. Please revise the text carefully.

Another point that I find irritating and that should be rephrased is the last sentence in the abstract ("From this study, we conclude that combining RF models with the RAMP monitors appears to be a very promising approach to address the poor performance that has plagued low cost air quality sensors.") and again on page 3 lines 1-3 ("as poor signal-to-noise ratios may hamper their ability to distinguish between intra-urban sites. As such, there has been increasing interest in more sophisticated algorithms (e.g., machine learning) for low cost sensor calibration."). These two statements are misleading as they imply that the limiting factor of sensor based data is data processing and not the gas sensing unit itself. It is well known that there are sensors available that are not sensitive and selective enough for the measurement of air pollutants at ambient concen-trations. Sophisticated algorithms will not be able to help here. The text should be changed so that the message of the paper is that sophisticated algorithms can improve the performance of those sensors that are generally suited for the measurement of ambient air pollutants.

On page 8, second paragraph it is stated that "The random forest model's main limitation is that its ability to predict new outcomes is limited to the range of the training dataset; in other words, it will not predict data with variable parameters outside the training range.". This is a relevant and important point and should further be discussed, i.e. the authors should elaborate on the practical consequences for using sensors. For example, the calibration model for O3 might not be applicable for peak summer concentrations when the training data has been measured during the cold season (how is the situation here, training data has been measured form August to February, is it applicable for peak ozone as typically observed in June/July?). This issue is even more

important for a multipollutant unit like the RAMP as pollutants like ozone have highest concentrations during summer and other primary pollutants often show highest concentrations during the cold season. Does this mean that calibration measurements need to cover a whole year, or what are the strategies for dealing with this situation?

The average Pearson correlation coefficients (e.g. the 0.99 for LAB and RF – even for CO2) are hardly to believe, given e.g. the scatter plots in Figure 4. There is a lot of scattering for all pollu-tants. On page 11 (line 5) the authors mention "The poor per-formance of linear models at predict-ing CO2 concentration is not surprising . . .". why then r=0.99 in Table 2? This needs to be checked or requires a convincing explanation. In addition, on page 11 line 31 it is said that "the Pearson r for NO2 ranged from 0.92 to 0.95". Again, this is very hard to believe, looking at Figure 5 there are a few RAMPs where I expect that r is smaller than 0.92 (e.g. #4, #6 #19). Please correct, or add the r values to the plots in Figure 5.

Other comments: The authors use alternately the terms "multivariate linear regression" and "multiple linear regres-sion". The method applied here is multiple linear regression and not multivariate linear regression which is something different. Use solely the term multiple linear regression.

On page 4, lines 20-21. The RAMP version with PM2.5 sensor does not need to be mentioned here since PM2.5 measurements are not used in the study. The notation of equations 1 and 2 is poor and should be improved. The measurements with the reference instruments are used in the models as independent variables, this should be clear. So use something like y_reference (t) = . . . instead of Corrected_MLR etc.

Page 8, line 22. The software package R should be correctly cited, see citation() in R.

Page 10, first paragraph. What is "the standard deviation of the model"? Is this the standard devi-ation of the model predictions? Please be clear and correct.

Page 12, line 8: "Smaller bias of RF models than the reference method?" Do you really

mean that the RF corrected sensor data have a smaller bias than the reference? How can this be, the reference measurements have been used as independent variable for training the RF models.

Page 14, line 9, it was found that the CO signal was the most important variable in the RF model for CO2. This likely poses strong limitations for using calibrated CO2 sensors in another environment than the location where the training data was obtained. The sensor calibration can likely not be transferred to rural environments, i.e. away from combustion sources, were CO and CO2 might not be strongly in-terlinked. What about measurements during the vegetation period, when CO2 uptake by plants can changes the relationship between CO2 and CO2 in urban environments? The authors should address this issue.

Legend of Figure 11 is wrong, should be RAMP vs. Reference, not the other way around.

Page 15, lines 24-26: "For NO2, the performance of 'out-of-the-box' low-cost sensors varied widely and half the sensors in the EuNetAir study (Borrego et al., 2016) reported errors larger than the average ambient concentrations. Therefore, advanced calibration models, such as those using machine learning, are critical to accurate measurements of ambient NO2.". As mentioned earlier, this is too simple and is neglecting the requirements for the gas sensing unit. If the sensor strongly responds to other factors than covered by the available predictors, not even advanced calibration models can be successful. So the quality of the sensing unit itself is key. The text should be revised.

Figure 12: More relevant than the slope and intercept of the regression of the RAMP against the reference would be the uncertainty of the daily 8h max as measured with the RAMP. This could be expressed by corresponding confidence intervals.

---

## Short Comment (SC1) · 4 Oct 2017

**Closing the gap on lower cost air quality monitoring: machine learning calibration models to improve low-cost sensor performance**

Review of Zimmerman et al.,

Eben S. Cross, Leah R. Williams, Gregory R. Magoon

Aerodyne Research, Inc. Billerica, MA 01821 USA

As written the Zimmerman et al., manuscript is an important contribution to the growing body of low-cost AQ sensor characterization efforts. The tone of the manuscript is a bit over-stated (incl. the title), as pointed out by the other reviewers, and the overall impact of the work could be improved if the authors more carefully addressed the following points of concern:

- Scope of work completed
    - The manuscript strongly emphasizes the unprecedented scale/scope of the completed work, stating that 19 RAMP systems were deployed for 6 months. At face value this would constitute a ~ 24wk interval across which to train & test the model. The actual reported tests appear more selective (both in terms of the number of RAMPS and duration of testing interval). As written, this is somewhat misleading. The authors should make an effort to more clearly state the scope of work as it pertains the results presented in the paper.
        - Pulling data reported in table 3:
        - CO: Test data spanned as few as 10 days, up to 108 days with an average of less than 6 weeks. Figure S7 shows only 16 of 19 RAMPS for evaluation (despite fact that 19 systems were RF-trained)
        - $NO_2$: Test data spanned as few as 2 days up 56 days with an average of 3.4 weeks. Figure S8 shows only 10 of 19 RAMPS were evaluated (despite fact that 19 systems were RF-trained)
        - $O_3$: Test data spanned 11-103 days (average less than 6 weeks) with 16 out of 19 system evaluated
        - $CO_2$ 15 out of 19 systems evaluated and the number of days of test data were not tabulated.
        - What is the fraction of training-to-test data for each RAMP system for which statistical metrics were reported?
        - Data displayed for RAMP #4 in Figure 8 shows 15 weeks of test data. From the average number of test sample days reported in Table 3, is RAMP #4 a significant outlier? Did the majority of other RAMP systems run for shorter periods of time?
    - While the authors point out that the limited $NO_2$ training/test data was due to a malfunction in their reference monitor at the co-location site, that does not explain why only 10 out of the 19 RAMP systems which were trained with the ambient RF model were included in the presented results.
        - The authors should comment on the impact of the significantly shorter evaluation period on the $NO_2$ results. Specifically, did the loss of the $NO_2$ reference monitor exclude data sampled over the colder or warmer

seasons in Pittsburgh and if so, how would this impact the range of conditions across which the RF model was found to be robust?

- Laboratory calibrations
  - o As the authors' correctly point out, laboratory calibrations have formed the basis for much of the low-cost AQ sensor characterization work completed to-date. The manner in which the laboratory calibration experiments were executed in the current work raises a number of concerns:
    - ▪ The authors should justify their laboratory calibration approach, specifically, sampling the sensors under 9 LPM of active flow, under air compositions dominated by (presumably) clean air, doped with single species of interest (excluding O3) under RH conditions that are outside of the specified operating range of the electrochemical sensors being trained. Given that these sensors operate under diffusion limited conditions, active vs passive flow can have a significant effect on the rate with which analyte molecules reach the working electrode surface of each electrochemical sensor. From the picture of the RAMP node, it appears that when fully integrated, the sensors are positioned to sample the air passively. This disconnect between the LAB cal. conditions and the ambient sampling configuration should be addressed if the authors are honestly trying to assess the validity of the LAB model on reconciling ambient concentrations from deployed RAMP monitors.
    - ▪ The lack of any systematic logging or control of temperature and RH under these laboratory conditions limits the overall usefulness (and relevance) of the laboratory calibration to reconciling ambient concentrations. While the LAB model is limited in its sophistication, the execution of the lab experiments themselves also presents environmental conditions that do not overlap with their ambient co-location conditions. This apparent disconnect between the LAB and field needs to be explained further.
    - ▪ The absence of any O3 lab calibrations needs to be explained further. Why was this species excluded and given the RF model assessment of the Ox-B431 sensor sensitivities to different parameters, do the authors think this sensor type would provide more reasonable LAB-based calibration models, if such experiments had been conducted?

- RF model
  - o With access to 1s reference monitor data it is not clear why the authors chose to use 15 min averages to train and test their RF model. Were shorter or longer time-averages tested and found to be measurably worse than the 15-min averages? What are the implications of using 15-min average data vs 1 or 5-min average data when resolving heterogeneity in local pollution gradients?

- o The authors should expand on their discussion regarding the lack of any extrapolation in the RF model.
  - ▪ (related) Figure 5. For RAMPS #9,12,13,18 the authors should explain the straight vertical and horizontal at the ~ (50,50) x,y position on each scatter plot.
- o It would be informative if the authors could comment on the computational cost of running the model. Does this computational cost place constraints on the time-averaging used to train the model in the first place?

- - P13 discussion of explanatory variables
  - o What do the authors mean by permuting? Replace with another dataset that's not related to the current dataset? A more thorough explanation of this process is warranted as this process appears critical to evaluating the importance of various interfering factors on each sensor type.
  - o Figure 9. Why is $CO_2$ more sensitive to CO than $CO_2$?
  - o The authors state that $SO_2$ concentrations were below detection limits for the duration of the ambient co-location study and therefore not discussed further in the manuscript. While it is true that the $SO_2$ concentrations in Pittsburgh are very low, the extent to which the SO2-B4 sensor output informed the RF model is in fact statistically significant according to the data presented in Figure 9 which indicates that the MSE can change by ~ 20-40% when the $SO_2$ sensor parameter (presumably differential voltage?) is permuted? A more robust assessment of the importance of the SO2-B4 sensor data to the resulting RF model may be to exclude it altogether from the available input parameters used to train the model.

- - All goodness of fit discussions relative to Cross et al., 2017 need to be revised according to the results published in the final accepted version of that manuscript.

Additional comments –

- - P11 L15: The figure caption does not indicate this…
  - o Figure 4 shows the calibrated RAMP #1 output regressed against the reference monitor concentration for the entire testing period for all three calibration models (LAB, MLR, and RF).
- - P12 L20: The text states that the MAE comparison is against the number of points, but Figure 9 displays this data versus the number of weeks, not number of points.
- - First paragraph of section 2.2 is unnecessarily repetitive
- - 95 sensor measurements (should be 76)..
- - P7 L6 'beta4' should be 'beta3' according to the formula above
- - P9 L20 missing 'resolution' following 'temporal'
- - P11 L13 Figure 2 should read Figure 3
- - P18 L7 missing 'this' - as written: 'demonstrate that degree'
- - P20 L30 Levy 2014 reference is the same as Moltchanov et al., 2015 reference.
- - Figure 2 caption should specify units as 'a.u.' following >255.9

- Figure 4 (left) – why do the four different pollutant times series all have unique time-periods?  If environmental parameters impact the sensors differently (RH, T) then it would be important to keep these parameters self-similar across the evaluation-framework presented here (even though it's only 48-hours worth, should be the same 48 hours for all sensors).
- Figure 7.  It's not clear why there are ~ 10 or fewer data points displayed when data from 19 RAMPS are reportedly presented
- Figure 8 caption.  'long periods' is relative.  Data displayed is for 15 weeks.  Lifetime of the sensors is significantly longer than this (~100-150 weeks).  Language should be revised accordingly.
  - o  The extent to which the model improves over time should be quantified with 95% confidence intervals on the linear fits.  By eye, it looks like this confidence interval would include 0.
- Table 3.  Rather than identifying the number of days of sampling/evaluation – it would be more appropriate to identify the total number of data points used in each case study.
  - o  Add an extra column that identifies the time resolution – as this is an important factor that drives signal-to-noise and accuracy and precision metrics as well as various end-use cases of interest.
- Section 4.4.  As written, this section oversimplifies the reality of the situation. When analyzing various lower-cost AQ sensor systems it is important to recognize that the combined hardware and software configuration impacts the performance metrics, not the software alone.  The authors shouldn't gloss over this fact.

---

## Author Comment (AC1) · 2 Nov 2017

**Response to Comments from Reviewer #1 AMT-2017-260**

The authors would first of all like to thank reviewer #1 for the insightful comments on the work we have submitted for publication, and the editor for the opportunity to improve the manuscript. Under each comment there is a summary of the response (red text), in addition to the text from the paper that was modified, if applicable.

**Reviewer #1**

The title is a bit ambitious, ambiguous, or both. How much of the performance "gap" is closed by a) improved hardware compared to past studies, b) the algorithm (i.e., Random Forest), c) sensor combinations at each node, and d) range of different sample types collected? Application of machine learning for sensor calibration in the field has been performed before, but the title and abstract seems to give the impression that this reduces the gap. There is much focus given to RF but there is no indication that it has an inherent advantage over other machine learning methods. For instance, it is possible that a MLR model could also handle cross-sensitivities only if it were provided all variables (though RF and other machine learning algorithms are more flexible in that it does not require the assumption regarding global linearity).

The past work of De Vito et al. (2008, 2009) also show encouraging results from a long-term evaluation of field calibrations (for low-cost multi-sensor devices for benzene, CO, and NO2 against government monitoring station instruments using machine learning algorithms).:

De Vito S., Massera E., Piga M., Martinotto L., and Di Francia G.: On field calibration of an electronic nose for benzene estimation in an urban pollution monitoring scenario, Sensors and Actuators B: Chemical, 129(2):750–757, doi:10.1016/j.snb.2007.09.060, 2008.

De Vito S., Piga M., Martinotto L., and Di Francia G.: CO, NO2 and NOx urban pollution monitoring with on-field calibrated electronic nose by automatic bayesian regularization, Sensors and Actuators B: Chemical, 143(1):182–191, doi:10.1016/j.snb.2009.08.041, 2009.

**Response:** Thank you for suggesting the papers by De Vito et al. and for correctly pointing out that the title is too bold. We have revised the title to: **"A machine learning calibration model to improve low-cost sensor performance".** We have also added references to De Vito et al. 2008 and 2009:

Modified text in Introduction (additions in bold):

"To date, there have been published studies using high-dimensional multi-response models (Cross et al., 2017) and neural networks (Esposito et al., 2016; Spinelle et al., 2015, 2017, **De Vito et al.**, **2008, 2009**). Spinelle et al. (2015) showed that artificial neural network calibration models could meet European data quality objectives for measuring ozone (uncertainty < 18 ppb); however, meeting these objectives for NO2 remained a challenge. In De Vito et al. (2009), the neural network calibration approach was applied to CO, NO2 and NOx metal oxide sensors in Italy with encouraging results; in general mean relative error was approximately 30%."

The manuscript is perhaps too bold in its tone. Accurate predictions are shown for concentration (and T, RH) domains that are present at the location of the reference monitor used for calibration,

even while using different data points. (As stated by the authors, current implementation of RF is limited to the domain of the training set.) Dense network coverage implies monitor placement in different microenvironments (e.g., nearroadway, etc.) which would experience different concentration regimes. Moreover, some of the explanatory variables used for calibration may be surrogates for another variable which may vary differently at another site. There is mention of two RAMPS units deployed in Pittsburgh and their positive evaluation against other reference measurements in a mobile van (p. 17, line 15), but no results are shown.

**Response:** Due to the currently long length of the manuscript, we have elected to not go into details of the mobile van measurements, and they will be presented in a forthcoming publication. However, we did deploy a RAMP that was calibrated at Carnegie Mellon University at the Allegheny County Health Department (ACHD) in February 2017-May 2017 and observed good agreement between the hourly ACHD concentrations of O3, NO2 and CO and the calibrated-RAMP. We have modified the manuscript to include this additional figure.

Below is the complete Section 4.5, which was re-organized to improve narrative flow and now includes the ACHD assessment (additions in bold), followed by the new Figure 12.

**"4.5 RF model calibrated RAMP performance in a monitoring context**

We further assess the RAMP monitor performance against three metrics: 1) comparison of a **RAMP monitor calibrated at Carnegie Mellon against an independent set of regulatory reference monitors at the Allegheny County Health Department**, 2) for NAAQS compliance, and 3) for suitability for exposure measurements as per the US EPA Air Sensor Guidebook (Williams et al., 2014). We also demonstrate the benefit of improved performance of the RF models in a real-world deployment at two nearby sites in Pittsburgh, PA.

From February through April 2017, a RAMP calibrated at the Carnegie Mellon Campus was deployed at the Allegheny County Health Department (ACHD) to test the performance of the RAMP relative to an independent reference monitor (Figure 12). The ACHD reports data hourly, so RAMP data were down-sampled to hourly averages and the CO, NO2 and O3 concentrations were compared (no measurement of CO2 is made at ACHD). For all pollutants,  $R^2$  was  $\geq 0.75$  (CO: 0.85, NO2: 0.75, O3: 0.92) and points were clustered around the 1:1 line. NO2 performed the most poorly, with a large cluster of points in the 5-10 ppb range where the model is known to underperform. The MAE was 49 ppb (17% CvMAE) for CO, 4.7 ppb for NO2 (39% CvMAE) and, 3.2 ppb for O3 (16% CvMAE), in line with the performance metrics in Figure 6.

[revised manuscript text omitted]

Since corrections of the supersite reference monitors against the Allegheny County Health Department instruments are necessary, why not make this Allegheny County Health Department site the reference site? Given the local contributions of vehicle emissions to CO and NO2 that are present in the parking lot site, how were the corrections for baseline drift determined?

**Response:** We have added two sentences to section 2.3 to describe the baseline correction approach. We would like to emphasize that the baseline corrections were modest and did not substantially affect the dataset from our reference monitors. The incentive for using the Carnegie Mellon site as the reference monitoring station is due to the higher time resolution of the data (we report at 1 Hz), the availability of the data in near-real time, and the ability to explore calibrations for pollutants not measured at the Allegheny County Health Department (ACHD) (e.g., CO2). Given the large numbers of RAMPs and availability of reference-grade instruments at CMU, the

CMU Supersite was much easier to access and hence used as the reference site. Other users who do not have the facilities we do could use their local regulatory monitors as a reference site if accessible.

Modified text in Section 2.3 (additions in bold)

"The CO and NO2 analyzers experience modest baseline drift between weekly calibrations, on the order of approximately 40 ppb for CO and 2 ppb for NO2. Hence, baseline pollutant concentrations were normalized to a nearby regulatory monitoring site (Allegheny County Health Department, Air Quality Division, Pittsburgh, PA). The baseline correction was done using a linear regression between the beginning and end of the week on the baseline signals (local source spikes removed). The regression was based on daytime differences, as night time inversions may cause real differences in the baseline signals between the two sites."

While the authors describe the use of 5-fold CV to selection the explanatory variables to use, the choice of 5 data points per terminal node / 100 trees per fold does not seem to be explained. This was also selected in the CV process?

**Response:** The typical range of cross-validations that are explored is from 3-20 folds. We observed that by 5 folds, the model performance had roughly stabilized, thus to optimize computational power we chose the minimum number of folds such that an increase in folds produced a <5% increase in model RMSE and R2. Similarly, random forests are typically constructed with 64-128 trees, so we chose a number in the middle of this range (100 trees). We agree that these details should be included in the manuscript, and have been added to Section 3.3.

Modified text in Section 3.3 (additions in bold):

"The number of trees was capped at 100 per fold, and a five-fold cross-validation was used for a total of 500 trees. Therefore, the predicted value for a given set of measured inputs is the average value from this set of 500 trees (each tree provides one prediction). The k-value was chosen by identifying the minimum number of folds for which an increase in the fold size increased model performance less than 5% on the held-out data. The number of trees was chosen based on the work of Oshiro et al. (2012), who suggested that the number of trees range from 64-128."

p. 14 Line 18 paragraph: Is this not possibly a limitation of the hardware?

**Response:** In this instance, we do not believe it is a limitation of the hardware. In our laboratory calibrations, we have exposed the sensors to several ppm of  $NO_2$  and have not observed a flat response (i.e., sensors are sensitive at high concentrations).

Minor comments:

Section 2.2: Data coverage (i.e., missing data) and the time resolution should be stated here rather than (or in addition to) later in the manuscript.

**Response:** Thank you for this comment that has also been pointed out by other reviewers. We have been more upfront with missing data and time resolution earlier in the manuscript to make the scope of the work clear.

Modified text in Section 2.2 (additions in bold)

"The experiments involved 95 individual pollutant sensors mounted in 19 unique RAMP monitors. While the collocation period spanned August 2016-February 2017, some sensors were intermittently deployed for air quality campaigns in Pittsburgh, thus the range of collocation available ranged from 30 days to the full collocation period, depending on the unit. Additionally, calibrations were not built for sensors for which reference data was below detection limits or if reference monitoring units were malfunctioning, reducing the total number of sensors in this experiment to 73, due to issues with the SO2 and NO2 monitors.

The electrochemical sensor outputs were measured using electronic circuitry custom designed by SenSevere optimized for signal stability. The circuitry includes custom electronics to drive the device, multiple stages of filtering circuitry for specific noise signatures, and an analog-to-digital converter for measurement of the conditioned signal. The RAMP monitors are housed in a NEMA-rated weather proof enclosure (Figure 1A) and equipped with GSM cards to transmit data using cellular networks to an online server. The RAMP monitors also log data to an SD card as a fail-safe in case of wireless data transfer issues. **The data is logged to the server at ~15 second resolution and down-sampled to 15-minute averages, which was deemed to be an appropriate time resolution for assessing spatial variability in air pollution exposure and to reduce the size of the dataset. Regulatory bodies typically make their data available at hourly resolution.**"

P. 9 Line 15 to end of paragraph. The authors switch from describing "intermittent" collocation to "distributed" collocation. Given the discussion of multiple RAMP monitors, "distributed" can be confusing. Also, "degree of collocation" is referring to frequency or effective duration?

**Response:** Thank you for pointing this out, we agree that it is confusing. We have switched the terminology to "consecutive" and "non-consecutive" collocations.

Modified Text in Section 3.3 (additions/changes in bold)

"This was evaluated for a consecutive collocation window and for 8 **non-consecutive** collocation windows equally distributed throughout the whole collocation period (August 2016 – February 2017) in half week increments. Details of this evaluation are provided in the Supplemental Information, but the **non-consecutive** collocations generally performed slightly better, with reductions in MAE of 12 ppb (4% relative error) for CO, 2 ppm for CO2 (0.4% relative error), 0.4 ppb for NO2 (4% relative error), and 1.6 ppb for O3 (7% relative error) compared to the consecutive four-week collocation. The motivation for exploring **non-consecutive** collocation windows dispersed throughout the study period was to ensure that the training period covered a complete range of gas species concentrations, temperatures and relative humidity. In practice, **the training data** utilized in this study is equivalent to collocating the RAMP monitors with reference monitors for 3-4 days every 1-2 months. **If non-consecutive collocation is inconvenient or not possible**,

**consecutive collocation may be satisfactory as determined by MAE and other accuracy parameters needed for the application at hand."**

p. 10 Line 19: value of correlation for NO2 and CO2 with reference monitors is missing.

**Response:** Thank you, this has been added.

Modified text in Section 4.1 (additions in bold):

"However, only the RF model achieved strong correlations between the reference monitor and the RAMPs for NO2 and CO2 (**Pearson r: 0.99**)."

p. 10 Line 22: insert figure numbers (SI Fig S3-S6).

**Response:** Thank you, this has been added.

Modified text in Section 4.1 (additions in bold):

"Regression plots for all 19 RAMPs and all four gas species illustrating the goodness of fit of the RF model are provided in the Supplemental Information (**Figures S3-S6**)."

p. 10 Line 30: The relationship between m\_try and model complexity is not very clear.

**Response:** We have edited Section 4.1 to add additional details to help make this connection clearer. In general, by having a larger m\_try, there is a higher probability that one dominant variable will be what the split is decided on. In other words, there is a lower probability that all the variables will participate in the model structure. If the model performance improves by diversifying the variables it splits based on, it is generally considered to have a more complex underlying structure. We have modified the text to better convey this point.

Modified Section 4.1 below (additions in bold)

"In general, the larger the mtry, the simpler the underlying structure of the model. For example, if there is one dominant variable but the model is permitted to consider all 7 explanatory variables at each decision node (i.e., mtry=7), then the model will most frequently split the data based on the dominant variable. **By contrast**, the advantage of a lower mtry is that subtle relationships between explanatory variables and the response can be probed. When randomly selecting fewer explanatory variables (mtry=2 or 4) at each decision node, the probability of selecting a dominant variable decreases and the model is forced to split the data into sub-nodes based on variables which may have a smaller (but real) effect on the response. If the goodness of fit of the calibration model is improved by decreasing mtry, this suggests more complex variable interactions with the response (Strobl et al., 2008)."

p. 11 Line 13: "clearly outperformed" -> not for CO

**Response:** As a general theme, we have toned down the language. We agree that for CO, any calibration seems to perform well and have modified the manuscript to reflect this.

Modified text in Section 4.2 (additions in bold, also removed the word "clearly":

"For this period, the RF model <del>clearly</del> outperformed the LAB and MLR models **for all pollutants except for CO**."

p. 11 Line 21: insert figure numbers (SI Figs S7-S10). Slopes, correlations, or some of the metrics listed in Table S2 included in the panels would be informative. Why are some RAMPS not included?

**Response**: Thank you, we acknowledge that why some RAMPs were not included was not totally clear, so we have made several revisions throughout the manuscript to be more descriptive of the calibration and collocation process. The total study domain was from August 2016 – February 2017, but RAMP monitors were intermittently deployed for air quality campaigns, so the average collocation period ranged from 5.5-15 weeks (median 9 weeks). After determining that 4 weeks of data was needed for proper calibration, some RAMP monitors did not have sufficient data to build a complete model (only 16 of the 19 RAMPs for NO2) and some did not have enough data for a meaningful testing period (minimum threshold 48 hrs, actual test window: 1.4-15.5 weeks). Thus for testing the model, the total number of RAMP monitors was reduced to 16 for CO and O3, 15 for CO2 and to 10 for NO2. We have modified the text in several sections to indicate this more clearly, with one example shown below. We have also added references to the Figure numbers in the text, and added the MAE and Pearson r metrics to the panels in Figures S7-S10, as requested (not showing here due to size of Figures, but is in Revised Manuscript).

Modified text (additions in bold) in Section 4.2:

"To assess the overall model performance, two performance metrics (Pearson r and CvMAE) were calculated for each RAMP monitor using the entire testing dataset (Figure 6). In this study, any data remaining after training were used to test model performance, provided there were at least 48 hours of testing data (192 data points). This reduced the number of RAMP monitors included for testing the model to 16 for CO and O3, 15 for CO2 and 10 for NO2. The size of the testing dataset varied from 1.4 to 15 weeks, with a median value of 5 weeks.

p. 11 Line 31: "NO2" -> "O3" here?

Response: Yes, thank you, that was a typographical error and has now been corrected.

---

## Author Comment (AC2) · 2 Nov 2017

**Response to Comments from Reviewer #2 AMT-2017-260**

The authors would first of all like to thank reviewer #2 for the insightful comments on the work we have submitted for publication, and the editor for the opportunity to improve the manuscript. Under each comment there is a summary of the response (red text), in addition to the text from the paper that was modified, if applicable.

**Reviewer #2**

The overall message of the manuscript is in my view too optimistic and can for readers be misleading. The authors should make clear that the good performance of the sensors found in this calibration study does not imply that the sensor unit is capable of providing similarly accurate air quality measurements in a real-world application. A good performance of sensor units in a calibration exercise like the study at hand is certainly necessary but not sufficient for the suitability of the sensors for real world air quality measurements. It should be clear that the manuscript is targeting on the good data quality obtained when combining the multi-pollutant sensor unit and RF and that a full assessment of the performance of the RAMPs within a sensor network for air quality measurements under real world conditions requires future research (and solutions for the quality assurance and quality control of the deployed sensors). The authors touch this point briefly in the conclusions section, however, for readers the impression remains that the RAMPS sensor units are ready for being used for urban air quality assessments. For example, in the conclusions section, last paragraph, it is stated that "Overall, we conclude that with careful data management and calibration using advanced machine learning models, that low-cost sensing with the RAMP monitors may significantly improve our ability to resolve spatial heterogeneity in air pollutant concentrations.". This conclusion is not justified by the available study and should be kept for the future work when results on the data quality as obtained in real world applications are available. As another example, the authors write on page 14, lines 14-16 "The US EPA limit of detection for federal regulatory monitors is 10 ppb for both NO2 and O3, suggesting that as with CO, the RF model performance is within 20% of regulatory standards (United States Environmental Protection Agency, 2014)". This is again misleading: It can be concluded from this calibration study that the performance of sensors with an updated calibration meet those requirements, the data quality that can be achieved with the sensor under real world conditions is something different and currently not known. Please revise the text carefully.

**Response:** We thank the reviewer for the comments on the manuscript. Respectfully, our calibration represents a real-world application tested under real-world conditions. The development and testing of the calibration occurred outdoors in an urban background environment with variable local sources such as passenger vehicles, trucks and restaurant emissions from nearby restaurants on the Carnegie Mellon campus. As such, there were real-world variants between the training and testing data. We do agree that the manuscript as written in the original submission only focused on testing data from RAMP monitors also at the Carnegie Mellon campus. To further demonstrate the suitability of the calibrated RAMP monitors in other real-world environments where traditional reference monitors were deployed, we have modified the manuscript to include a comparison of a RAMP monitor calibrated at Carnegie Mellon and then moved to the Allegheny County Health Department (ACHD), where there is an independent set of reference monitors for

CO, NO$_2$ and O$_3$. The ACHD site has more nearby sources than the Carnegie Mellon site (more traffic and restaurants) and different land use classifications. Comparing the CMU calibrated RAMP to the ACHD data, we found good agreement for the pollutants at similar performance levels (based on CvMAE, Pearson r) to the testing data originally presented in the manuscript. These results are included in a new Figure 14 with additional text. We have also added additional wording to Section 2.2 to indicate the nature of the real-world environments tested as part of this study.

Additional text in Section 4.5 (new text in bold), followed by the new Figure 12 (old Figures 12-14 now shifted by one):

"We further assess the RAMP monitor performance against **three** metrics: **1) comparison of a RAMP monitor calibrated at Carnegie Mellon against an independent set of regulatory reference monitors at the Allegheny County Health Department**, 2) for NAAQS compliance, and 3) for suitability for exposure measurements as per the US EPA Air Sensor Guidebook (Williams et al., 2014). We also demonstrate the benefit of improved performance of the RF models in a real-world deployment at two nearby sites in Pittsburgh, PA.

**From February through April 2017, a RAMP calibrated at the Carnegie Mellon Campus was deployed at the Allegheny County Health Department (ACHD) to test the performance of the RAMP relative to an independent reference monitor (Figure 12). The ACHD reports data hourly, so RAMP data were down-sampled to hourly averages and the CO, NO2 and O3 concentrations were compared (no measurement of CO2 is made at ACHD). For all pollutants, R2 was ≥0.75 (CO: 0.85, NO2: 0.75, O3: 0.92) and points were clustered around the 1:1 line. NO2 performed the most poorly, with a large cluster of points in the 5-10 ppb range where the model is known to underperform. The MAE was 49 ppb (17% CvMAE) for CO, 4.7 ppb for NO2 (39% CvMAE) and, 3.2 ppb for O3 (16% CvMAE), in line with the performance metrics in Figure 6.**"

Additional description of the ACHD site in Section 2.1:

"**The RAMP monitors have also been intermittently deployed across the Pittsburgh region as part of ongoing air quality monitoring research. To demonstrate the accuracy of the calibrated RAMP, we also show data from a RAMP monitor which was first calibrated at Carnegie Mellon University and then moved to the Allegheny County Health Department (ACHD, 40°27'55.6"N, 79°57'38.9"W) from February – May 2017. The ACHD site has independent reference monitors for CO, NO$_2$ and O$_3$ and thus comparing data from these two sites enables an independent real-world assessment of model performance. The ACHD site is characterized by increased traffic volume, restaurant density and industry relative to the Carnegie Mellon site.**"

[Figure]

**Figure 12: Comparison of CO, NO₂ and O₃ hourly average concentrations measured by a co-located RAMP monitor and the reference monitors at the Allegheny County Health Department (ACHD). The RAMP monitor was first calibrated on the Carnegie Mellon campus prior to deployment.**

Another point that I find irritating and that should be rephrased is the last sentence in the abstract ("From this study, we conclude that combining RF models with the RAMP monitors appears to be a very promising approach to address the poor performance that has plagued low cost air quality sensors.") and again on page 3 lines 1-3 ("as poor signal-to-noise ratios may hamper their ability to distinguish between intra-urban sites. As such, there has been increasing interest in more sophisticated algorithms (e.g., machine learning) for low cost sensor calibration."). These two statements are misleading as they imply that the limiting factor of sensor based data is data processing and not the gas sensing unit itself. It is well known that there are sensors available that are not sensitive and selective enough for the measurement of air pollutants at ambient concentrations. Sophisticated algorithms will not be able to help here. The text should be changed so that the message of the paper is that sophisticated algorithms can improve the performance of those sensors that are generally suited for the measurement of ambient air pollutants.

**Response**: We agree that there are some gas sensing units that will never be suitable for air quality measurement applications and we have modified the text to more directly to address the numerous limiting factors for low-cost sensors. In the abstract, we only make this claim regarding our specific

unit (which is suited to measurement of ambient air pollutants), and not all gas sensing units, thus we have left the abstract unchanged.

Modified text in introduction:

**"The two primary requirements of low cost sensors for ambient measurement are 1) hardware that is sensitive to ambient pollutant concentrations, and 2) calibration of the sensors. The latter is the focus of this study. A primary challenge of low-cost sensor calibration** is that the sensors are prone to cross-sensitivities with other ambient pollutants (Bart et al., 2014; Cross et al., 2017; Masson et al., 2015b; Mead et al., 2013)"

On page 8, second paragraph it is stated that "The random forest model's main limitation is that its ability to predict new outcomes is limited to the range of the training dataset; in other words, it will not predict data with variable parameters outside the training range.". This is a relevant and important point and should further be discussed, i.e. the authors should elaborate on the practical consequences for using sensors. For example, the calibration model for O3 might not be applicable for peak summer concentrations when the training data has been measured during the cold season (how is the situation here, training data has been measured form August to February, is it applicable for peak ozone as typically observed in June/July?). This issue is even more important for a multipollutant unit like the RAMP as pollutants like ozone have highest concentrations during summer and other primary pollutants often show highest concentrations during the cold season. Does this mean that calibration measurements need to cover a whole year, or what are the strategies for dealing with this situation?

**Response:** We agree that this is a critical point to further emphasize. We have changed some of the language and added additional text on a possible solution for extrapolating, such as a hybrid RF and MLR model, where the MLR model is used for concentrations beyond the 95th percentile of the training data.

Modified text in Section 3.3 (additions in bold)

"The random forest model's **critical** limitation is that its ability to predict new outcomes is limited to the range of the training data set; in other words, it will not predict data with variable parameters outside the training range (no extrapolation). Therefore, a larger and more variable training data set should create a better final model. **In this study, our collocation window covered a broad range of concentrations and meteorological conditions; however, in situations where shorter collocation windows with less diverse training ranges are desired, the RF model may not be suitable as a standalone model. This is discussed further in Section 4.3.2.**"

Modified text in Section 4.3.2 (additions in bold)

"To build a robust model, many data points are required at a given concentration to probe the extent of the ambient air pollutant matrix. In this study, the training windows were dispersed throughout the collocation period to ensure good agreement of gas species and meteorological conditions during both the training and testing windows (see Supplemental Information). **The RF model may not work well in cases where such a diverse collocation window is not possible or where concentrations are routinely expected to exceed the training window. In such**

**situations, hybrid calibration models such as combined RF-MLR where MLR is used for concentrations above the training window range may be suitable, as MLR tends to perform better when concentrations are higher.**"

The average Pearson correlation coefficients (e.g. the 0.99 for LAB and RF – even for CO2) are hardly to believe, given e.g. the scatter plots in Figure 4. There is a lot of scattering for all pollutants. On page 11 (line 5) the authors mention "The poor performance of linear models at predicting CO2 concentration is not surprising . . .". why then r=0.99 in Table 2? This needs to be checked or requires a convincing explanation. In addition, on page 11 line 31 it is said that "the Pearson r for NO2 ranged from 0.92 to 0.95". Again, this is very hard to believe, looking at Figure 5 there are a few RAMPs where I expect that r is smaller than 0.92 (e.g. #4, #6 #19). Please correct, or add the r values to the plots in Figure 5.

**Response:** The numbers in Table 2 correspond to the goodness of fit of the model (i.e., performance of the withheld folds in the training data). As such, the scatter plots in Figure 5 are not related to Table 2 – but to the scatter plots in the Supporting Information (SI Figures S3-S6). The data shown in Figure 5 is for testing data (Section 4.2) We have added references to the SI figures in the text directly to minimize confusion, and modified the caption of Table 2 to direct the reader to Section 4.1 (discussion of goodness of fit). Additionally, as pointed out by reviewer number 1, there was a typo in the manuscript, the statement that the Pearson r varied from 0.92-0.95 is for $O_3$, not for $NO_2$ and was a simple typographical error that has been corrected.

Other comments: The authors use alternately the terms "multivariate linear regression" and "multiple linear regression". The method applied here is multiple linear regression and not multivariate linear regression which is something different. Use solely the term multiple linear regression.

**Response:** Thank you for noticing this – we have corrected all instances of "multivariate linear regression" to "multiple linear regression", these were typographical errors.

On page 4, lines 20-21. The RAMP version with PM2.5 sensor does not need to be mentioned here since PM2.5 measurements are not used in the study. The notation of equations 1 and 2 is poor and should be improved. The measurements with the reference instruments are used in the models as independent variables, this should be clear. So use something like y_reference (t) = . . . instead of Corrected_MLR etc.

**Response:** The two instances where PM2.5 are mentioned in Section 2.2 have been removed, as Reviewer #2 is correct that they are not used in the study. We have also revised the notation of Equations 1 and 2

Modified Equations 1 and 2 below:

$$y_{reference}(t) = \beta_0 + \beta_1 \times [\text{Net Sensor Response (CO, NO}_2) \text{ or Raw Sensor Response (CO}_2)], \quad (1)$$

$$y_{reference}(t) = \beta_0 + \beta_1 \times [\text{Net Sensor Resp. (CO, NO}_2, O_3) \text{ or Raw Sensor Resp. (CO}_2)] + \beta_2 \times T + \beta_3 \times RH, \quad (2)$$

Page 8, line 22. The software package R should be correctly cited, see citation() in R.

**Response:** We agree that the correct way to cite an R package is using citation() in R, and this is how the citation was generated. There appeared to be an issue translating the BibTeX file into the document, which we have now resolved.

Modified citation:

"Kuhn, M., Contributions from: Wing, J., Weston, S., Williams, A., Keefer, C., Engelhardt, A., Cooper, T., Mayer, Z., Kenkel, B., The R Core Team, Benesty, M., Lescarbeau, R., Ziem, A., Scrucca, L., Tang, Y., Candan, C. and Hunt., T.: caret: Classification and Regression Training, [online] Available from: https://cran.r-project.org/package=caret, R package version 6.0-76, 2017."

Page 10, first paragraph. What is "the standard deviation of the model"? Is this the standard deviation of the model predictions? Please be clear and correct.

**Response:** The reviewer is correct, we mean the standard deviation of the model predictions. We have modified the text accordingly.

Modified text (additions in bold)

"Since CRMSE is always positive, a further dimension is added: if the standard deviation of the model **predictions (calibrated sensor data)** exceeds the standard deviation of the reference measurements, the CRMSE is plotted in the right quadrants and vice versa. To match previously constructed target diagrams (Borrego et al., 2016; Spinelle et al., 2015, 2017), the CRMSE and MBE were normalized by the standard deviation of the reference measurements, and thus the vector distance in our diagrams is RMSE/$\sigma$reference (nRMSE). The resulting diagram enables visualization of four diagnostic measures: (1) whether the model tends to overestimate (MBE $>$ 0) or underestimate (MBE $<$ 0), (2) whether the standard deviation of the model **predictions (calibrated sensor data)** is larger (right plane) or smaller (left plane) than the standard deviation of the reference measurements, …"

Page 12, line 8: "Smaller bias of RF models than the reference method?" Do you really mean that the RF corrected sensor data have a smaller bias than the reference? How can this be, the reference measurements have been used as independent variable for training the RF models.

**Response:** As written, the manuscript states the RF model responses were "biased slightly lower" than the reference measurements, which we mean as "tend to underpredict" (negative MBE). This is not the same as saying the RF calibrated sensor data has less bias than the reference monitors, which we agree is not possible. We have rephrased to make this clearer.

Modified text in Section 4.2 (additions in bold):

"Across all gases, the RF models on average were **biased towards predicting** concentrations slightly lower than the reference **(i.e., slight tendency to underpredict, MBE/$\sigma_{reference}$ <0)**."

Page 14, line 9, it was found that the CO signal was the most important variable in the RF model for CO2. This likely poses strong limitations for using calibrated CO2 sensors in another environment than the location where the training data was obtained. The sensor calibration can

likely not be transferred to rural environments, i.e. away from combustion sources, were CO and CO2 might not be strongly interlinked. What about measurements during the vegetation period, when CO2 uptake by plants can changes the relationship between CO2 and CO2 in urban environments? The authors should address this issue.

**Response:** We agree that given the dependence of the $CO_2$ calibration on the CO signal that the sensors would likely not be suited for rural environments. We have added additional text to the manuscript to emphasize that these models would likely only perform best in urban environments unless a custom calibration was built in a rural environment.

Modified text in Section 4.3.1 (additions in bold):

"The explanatory variable importance is more complex for $CO_2$ and $NO_2$. For $CO_2$, all variables are important roughly equally important, with CO being the most important. This is likely due to the strong meteorological effect of humidity on the measured $CO_2$ concentration; the model must rely on other primary pollutants to predict CO2 signal when the measured $CO_2$ has reached full-scale, and short-term fluctuations of $CO_2$ are likely from combustion sources (e.g., vehicular traffic in urban areas) which also emit CO. This highlights the value of having sensors for multiple pollutants in the same monitor. Including measurements of additional pollutants helps the RF model correct for cross-sensitivities. **However, the drawback of this cross-sensitivity in the model is that the RF model may not perform well in areas where the characteristic source ratios of CO and $CO_2$ have changed. For example, this model was calibrated in an urban environment with many traffic and combustion-related sources nearby. Such a model would be expected to perform poorly for $CO_2$ in a heavily vegetated rural environment where CO and $CO_2$ are not strongly linked**."

Legend of Figure 11 is wrong, should be RAMP vs. Reference, not the other way around.

**Response:** Thank you for noticing this, the caption of Figure 11 has been corrected.

Modified caption (changes in bold):

Figure 11: Illustrating the range of predictions from the 500 trees for RAMP #1. The testing data were binned and averaged. The concentration measured by the  **calibrated RAMP monitors** is then plotted against the average concentration from the  **reference monitor**. The error bars represent the standard deviation of the answers from the 500 trees and the bins are colour coded by the number of data points within each bin. The dashed black line is the 1:1 line.

Page 15, lines 24-26: "For NO2, the performance of 'out-of-the-box' low-cost sensors varied widely and half the sensors in the EuNetAir study (Borrego et al., 2016) reported errors larger than the average ambient concentrations. Therefore, advanced calibration models, such as those using machine learning, are critical to accurate measurements of ambient NO2.". As mentioned earlier, this is too simple and is neglecting the requirements for the gas sensing unit. If the sensor strongly responds to other factors than covered by the available predictors, not even advanced calibration models can be successful. So the quality of the sensing unit itself is key. The text should be revised.

**Response:** We agree with the Reviewer's assessment that calibration alone cannot correct all sensors and we have modified the text to emphasize the complexity of the problem

Modified text in Section 4.4 is below (additions in bold):

For $NO_2$, the performance of 'out-of-the-box' low-cost sensors varied widely and half the sensors in the EuNetAir study (Borrego et al., 2016) reported errors larger than the average ambient concentrations. **While the quality of the baseline gas sensing unit remains critical (in which case no calibration should work), we suggest that advanced calibration models, such as those using machine learning, may be critical for accurate measurements of ambient $NO_2$.**"

Figure 12: More relevant than the slope and intercept of the regression of the RAMP against the reference would be the uncertainty of the daily 8h max as measured with the RAMP. This could be expressed by corresponding confidence intervals.

**Response:** We have added 95% confidence intervals to the MLR and RF-calibrated RAMP monitor daily 8h max as compared to the reference monitors to incorporate this uncertainty.

Modified Text in Section 4.5 (additions in bold):

"For the representative RAMP monitor used previously (RAMP #1), daily maximum 8-hour $O_3$ was in good agreement between the RF calibrated RAMP and the reference monitor, with all data points falling roughly along the 1:1 line (slope: 0.82, **95% CI: 0.81-0.83**), while for the MLR model, concentrations were skewed slightly low (slope of 0.65, **95% CI: 0.63-0.67**)."

---

## Author Comment (AC3) · 2 Nov 2017

**Response to Comments from Aerodyne Research Inc: AMT-2017-260**

The authors would first of all like to thank Aerodyne (and specifically Dr. Eben Cross) for the insightful comments on the work we have submitted for publication, and the editor for the opportunity to improve the manuscript. Under each comment there is a summary of the response (red text), in addition to the text from the paper that was modified, if applicable.

**Response to Comment from Aerodyne Research Inc. (Eben Cross et al.)**

1) Scope of work completed:

   The manuscript strongly emphasizes the unprecedented scale/scope of the completed work, stating that 19 RAMP systems were deployed for 6 months. At face value this would constitute a ~ 24wk interval across which to train & test the model. The actual reported tests appear more selective (both in terms of the number of RAMPS and duration of testing interval). As written, this is somewhat misleading. The authors should make an effort to more clearly state the scope of work as it pertains the results presented in the paper.

   o Pulling data reported in table 3:
   o CO: Test data spanned as few as 10 days, up to 108 days with an average of less than 6 weeks. Figure S7 shows only 16 of 19 RAMPS for evaluation (despite fact that 19 systems were RF-trained)
   o NO2: Test data spanned as few as 2 days up 56 days with an average of 3.4 weeks. Figure S8 shows only 10 of 19 RAMPS were evaluated (despite fact that 19 systems were RF-trained)
   o O3: Test data spanned 11-103 days (average less than 6 weeks) with 16 out of 19 system evaluated
   o CO2 15 out of 19 systems evaluated and the number of days of test data were not tabulated.
   o What is the fraction of training-to-test data for each RAMP system for which statistical metrics were reported?
   o Data displayed for RAMP #4 in Figure 8 shows 15 weeks of test data. From the average number of test sample days reported in Table 3, is RAMP #4 a significant outlier? Did the majority of other RAMP systems run for shorter periods of time?

   **Response:** We agree that the manuscript could have been more direct in terms of actual training and testing windows – the reason that longer collocations were not possible was due to deploying the RAMP monitors intermittently as part of air quality research campaigns in Pittsburgh, PA. While RAMP #4 did run the longest (was permanently located at the Carnegie Mellon supersite), there were 5 other RAMP monitors for which the testing period was 8-10 weeks. Additionally, we used RAMP #4 to systematically assess the performance of the testing data in Figure 8, and did not observe any significant relationship between weeks of testing data and error metrics up to 15 weeks. We only tested the models if there were at least 48 hrs of collocation data left after training; for 3 of the RAMPs, there was not enough data to properly test the model for all pollutants since it was deployed in the field. For an additional three RAMPs, there was not enough data to test the NO2 model due to the reference monitor being offline and needing repair from the manufacturer.

We have added additional language throughout the manuscript to more specifically address the specific training and testing windows.

Modified text in Introduction:

"To ensure calibration model robustness, they were developed for **16-19 RAMP monitors and validated for 10-16 RAMP monitors (depending on pollutant)**, with each monitor containing one sensor per species (CO, $CO_2$, $NO_2$, $SO_2$ and $O_3$). Furthermore, the study was conducted over a six-month period (August 2016 – February 2017) spanning multiple seasons and a wide range of meteorological conditions. **During this period, RAMP monitors were intermittently deployed for air quality monitoring campaigns, resulting in collocation periods ranging from 5.5 to 16 weeks (median 9 weeks).**"

Additional text in Section 2.2 (new text in bold):

"The experiments involved 95 individual pollutant sensors mounted in 19 unique RAMP monitors. **While the collocation period spanned August 2016-February 2017, many sensors were intermittently deployed for air quality campaigns in Pittsburgh, so the collocation period ranged from 30 days to the study period, depending on the unit. Additionally, calibrations were not built for sensors for which reference data was below detection limits or if reference monitoring units were malfunctioning, reducing the total number of sensors in this experiment to 73, due to issues with the $SO_2$ and $NO_2$ reference monitors.**"

Modified text in Section 4.1 (additional text in bold):

Regression plots for 19 RAMP monitors **for CO, $CO_2$ and $O_3$ and 16 RAMP monitors for $NO_2$** illustrating the goodness of fit of the RF model are provided in the Supplemental Information **(Figures S3-S6). Only 16 of the 19 RAMP monitors had an $NO_2$ calibration, since the $NO_2$ monitor malfunctioned during the period when three RAMPs were collocated and so a calibration model could not be built for $NO_2$ for these three RAMPs. The $NO_2$ malfunction occurred between late September and early October, which did not significantly impact the range of conditions across the study.**

Modified text in Section 4.2 (additional text in bold):

"To assess the overall model performance, two performance metrics (Pearson r and CvMAE) were calculated for each RAMP monitor using the entire testing dataset (Figure 6). **In this study, any data remaining after training were used to test model performance, provided there were at least 48 hours of testing data (192 data points). This reduced the number of RAMP monitors included for testing the model to 16 for CO and $O_3$, 15 for $CO_2$ and 10 for $NO_2$.**"'

2) While the authors point out that the limited NO2 training/test data was due to a malfunction in their reference monitor at the co-location site, that does not explain why only 10 out of the 19 RAMP systems which were trained with the ambient RF model were included in the presented results.

a. The authors should comment on the impact of the significantly shorter evaluation period on the NO2 results. Specifically, did the loss of the NO2 reference monitor exclude data sampled over the colder or warmer seasons in Pittsburgh and if so, how would this impact the range of conditions across which the RF model was found to be robust?

**Response:** As noted in the response to the previous comment, only 10 RAMP monitors were tested due to insufficient data available (i.e., as soon as those models had data to train the model, they were deployed for air quality monitoring). We have been more explicit about this in the text (see response to previous comment). Additionally, the $NO_2$ monitor experienced issues in late September-early October, which did not affect the range of $NO_2$ sensors for training.

Modified text in Section 4.1 (additional text in bold):

Regression plots for 19 RAMP monitors **for CO, CO₂ and O₃ and 16 RAMP monitors for NO₂** illustrating the goodness of fit of the RF model are provided in the Supplemental Information **(Figures S3-S6). Only 16 of the 19 RAMP monitors had an NO₂ calibration, since the NO₂ monitor malfunctioned during the period when three RAMPs were collocated and so a calibration model could not be built for NO₂ for these three RAMPs. The NO₂ malfunction occurred between late September and early October, which did not significantly impact the range of conditions across the study.**

3) Laboratory calibrations

As the authors' correctly point out, laboratory calibrations have formed the basis for much of the low-cost AQ sensor characterization work completed to-date. The manner in which the laboratory calibration experiments were executed in the current work raises a number of concerns:

o The authors should justify their laboratory calibration approach, specifically, sampling the sensors under 9 LPM of active flow, under air compositions dominated by (presumably) clean air, doped with single species of interest (excluding O3) under RH conditions that are outside of the specified operating range of the electrochemical sensors being trained. Given that these sensors operate under diffusion limited conditions, active vs passive flow can have a significant effect on the rate with which analyte molecules reach the working electrode surface of each electrochemical sensor. From the picture of the RAMP node, it appears that when fully integrated, the sensors are positioned to sample the air passively. This disconnect between the LAB cal. conditions and the ambient sampling configuration should be addressed if the authors are honestly trying to assess the validity of the LAB model on reconciling ambient concentrations from deployed RAMP monitors.

**Response:** The design of the sampling manifold was such that the face velocity at the sensor surface would be 1.2 m/s, which is in lower end of wind speed range in Pittsburgh (e.g. average monthly windspeed from Jan-May 2017 was 2.4-3.4 m/s). The gas flow rate for the calibration system was based on the required flow rate for the reference instruments, the need to avoid leaks of ambient air into the system, and to minimize calibration gas consumption. Additionally, each data point was taken after 20 min when gas concentrations

had stabilized as seen in the steady gas sensor output voltage. We have added these details to the manuscript, and changed the terminology from flow rate to face velocity for clarity.

Modified text in Section 3.1 (additions in bold):
"The sensors were exposed to each step in the calibration window (Table 1) for 20 minutes and a **face velocity of 1.2 m/s** flowed perpendicular to the sensor surface. **This face velocity is in the lower end of the wind speed range in Pittsburgh, PA (e.g. average monthly windspeed over Jan-May 2017 at 2m height is estimated at 2.4-3.4 m/s).**"

o The lack of any systematic logging or control of temperature and RH under these laboratory conditions limits the overall usefulness (and relevance) of the laboratory calibration to reconciling ambient concentrations. While the LAB model is limited in its sophistication, the execution of the lab experiments themselves also presents environmental conditions that do not overlap with their ambient co-location conditions. This apparent disconnect between the LAB and field needs to be explained further.
**Response:** We agree; however, our laboratory calibration was limited by the available infrastructure at the time of the study. The goal of the laboratory calibration was to quantify the correlation between analyte response and calibration concentrations. While the utility of the calibration is limited, it was also useful to know that the CO calibration performed well even with a simple linear model. We have added text to Section 3.1 to emphasize that the laboratory calibration could be improved and better performance is in theory possible.

Modified text in Section 3.1 (additions in bold):
"Model performance was evaluated by comparing the calibrated response to reference measurements. We refer to the laboratory univariate linear regression calibration as LAB. Separate LAB calibrations were developed for each sensor (95 individual calibrations). **Due to difficulty controlling temperature and RH over a wide range of known ambient conditions, we focused on the relationship between analyte response and the calibration gas concentration, which any user with access to basic lab infrastructure can do. While beyond the scope of this study, an improved LAB calibration would involve a chamber with variable T and RH to better match ambient conditions.**"

o The absence of any O3 lab calibrations needs to be explained further. Why was this species excluded and given the RF model assessment of the Ox-B431 sensor sensitivities to different parameters, do the authors think this sensor type would provide more reasonable LAB-based calibration models, if such experiments had been conducted?
**Response:** We did not conduct a LAB calibration for ozone due to our lack of a controlled low-concentration ozone generator and we did not find suitable ozone calibration gas. We cannot comment on the outcome of a LAB based calibration for O3 as no such experiments were possible. From the RF model, RH and T seemed to have minimal impact on the ozone calibration, but this would require further investigation.

Modified text in Section 3.1 (additions in bold):

"Laboratory calibrations for $O_3$ were not performed **due to lack of a suitable ozone calibration gas**."

4) RF Model
With access to 1s reference monitor data it is not clear why the authors chose to use 15 min averages to train and test their RF model. Were shorter or longer time-averages tested and found to be measurably worse than the 15-min averages? What are the implications of using 15-min average data vs 1 or 5-min average data when resolving heterogeneity in local pollution gradients?
**Response:** The raw RAMP monitor data is reported at 15 second intervals, and down-averaged to 15 minutes for two primary reasons: 1) the goal of the RAMP monitor deployment in Pittsburgh is to quantify long term spatial and temporal variability in air pollution for exposures and 2) to generate a manageable data set for when 50+ monitors are deployed in Pittsburgh.

Modified text in Section 2.2 (additions in bold):
"The RAMP monitors also log data to an SD card as a fail-safe in case of wireless data transfer issues. **The data is logged to the server at ~15 second resolution and down-sampled to 15-minute averages, which was deemed to be an appropriate time resolution for assessing spatial variability in air pollution exposure and to reduce the size of the dataset and increase computational efficiency. Regulatory bodies typically make their data available at hourly resolution.**"

The authors should expand on their discussion regarding the lack of any extrapolation in the RF model.
   o (related) Figure 5. For RAMPS #9,12,13,18 the authors should explain the straight vertical and horizontal at the ~ (50,50) x,y position on each scatter plot.

**Response:** This point was made by other reviewers and extensive changes have been made to emphasize this fact (see below). We have also noted that the horizontal features in Figure 5 are a result of the model being unable to extrapolate.

Modified text in Section 3.3 (additions in bold)

"The random forest model's **critical** limitation is that its ability to predict new outcomes is limited to the range of the training data set; in other words, it will not predict data with variable parameters outside the training range (no extrapolation). Therefore, a larger and more variable training data set should create a better final model. **In this study, our collocation window covered a broad range of concentrations and meteorological conditions; however, in situations where shorter collocation windows with less diverse training ranges are desired, the RF model may not be suitable as a standalone model. This is discussed further in Section 4.3.2.**"

Modified text in Section 4.3.2 (additions in bold)

"To build a robust model, many data points are required at a given concentration to probe the extent of the ambient air pollutant matrix. In this study, the training windows were dispersed throughout the collocation period to ensure good agreement of gas species and meteorological

conditions during both the training and testing windows (see Supplemental Information). **The RF model may not work well in cases where such a diverse collocation window is not possible or where concentrations are routinely expected to exceed the training window. In such situations, hybrid calibration models such as combined RF-MLR where MLR is used for concentrations above the training window range may be suitable, as MLR tends to perform better when concentrations are higher.**"

Modified text in Section 4.3.3 (additions in bold)

"Systematic underprediction at the highest concentrations was also observed and is likely a consequence of either sensor limitations or the training dataset used to fit the RF model. Unless the range of concentrations in the training data encompasses the range of concentrations during model testing, there will be underpredictions for concentrations in exceedance of the training range **due to the RF model's inability to extrapolate. This is also what causes the horizontal feature for some RAMP monitors at high $O_3$ concentrations in Figure 5, as the model will not predict beyond its training range.**"

It would be informative if the authors could comment on the computational cost of running the model. Does this computational cost place constraints on the time averaging used to train the model in the first place?

**Response:** We have added a comment on the computational cost of the model. The reviewer is correct that increasing the time resolution would come at a significant computational cost, as each RAMP monitor takes approximately 45 minutes to train at 15 minute resolution, thus when building calibrations for up to 50 RAMP monitors (ultimate goal of the work), increase time resolution could be prohibitive computationally.

Additional text in Section 3.3:

**"The computation time to train a complete RAMP monitor with five sensors was approximately 45 minutes. This was another motivating factor for 15 minute resolution data, as building models at higher time resolutions would have significantly increased computational demand**."

5) P13 discussion of explanatory variables

What do the authors mean by permuting? Replace with another dataset that's not related to the current dataset? A more thorough explanation of this process is warranted as this process appears critical to evaluating the importance of various interfering factors on each sensor type.

**Response:** The term permuting is a mathematical term which means that the signal is randomly shuffled, which is not the same as replacing with another dataset not related to the current data set. Both within the manuscript and within the figure caption for explanatory variable performance we describe permuting by saying "(i.e., randomly shuffled)" and thus we feel that the explanation as

offered in the manuscript and the standard nature of this mathematical concept does not warrant an expanded discussion.

Figure 9. Why is CO2 more sensitive to CO than CO2?
**Response:** In periods of high humidity, the $CO_2$ sensor becomes saturated, as the NDIR $CO_2$ sensor is also sensitive to water. We hypothesize that when the $CO_2$ sensor saturates, the model must rely on other pollutant signals (e.g., CO) as a predictor of $CO_2$ concentration. Additionally, short term fluctuations of CO2 are likely from combustion sources which also emit CO. This is currently in the manuscript in Section 4.3.2, but we have also added a few words of clarifying text.

Text from Section 4.3.2 (additions in bold):
"For $CO_2$, all variables are important roughly equally important, with CO being the most important. This is likely due to the strong meteorological effect of humidity on the measured $CO_2$ concentration; the model must rely on other primary pollutants to predict CO2 signal when the measured $CO_2$ has reached full-scale **(i.e., becomes saturated in periods of high humidity)**, and short-term fluctuations of $CO_2$ are likely from combustion sources (e.g., vehicular traffic in urban areas) which also emit CO. This highlights the value of having sensors for multiple pollutants in the same monitor.

The authors state that SO2 concentrations were below detection limits for the duration of the ambient co-location study and therefore not discussed further in the manuscript. While it is true that the SO2 concentrations in Pittsburgh are very low, the extent to which the SO2-B4 sensor output informed the RF model is in fact statistically significant according to the data presented in Figure 9 which indicates that the MSE can change by ~ 20-40% when the SO2 sensor parameter (presumably differential voltage?) is permuted? A more robust assessment of the importance of the SO2-B4 sensor data to the resulting RF model may be to exclude it altogether from the available input parameters used to train the model.

**Response:** The SO2 RAMP sensor may in and of itself have useful cross-sensitivities that may assist model performance. This is likely why the RAMP $SO_2$ sensor contributes to model performance despite low ambient $SO_2$ concentrations, thus we elected to include it. We have added some text regarding this hypothesis to Section 4.3.2.

Modified text in Section 4.3.2 (additions in bold):
"**Interestingly, despite low $SO_2$ concentrations, there was some contribution from the RAMP $SO_2$ sensor. This may be due to cross-sensitivities within the $SO_2$ sensor itself, as the $SO_2$ sensor may respond to more than ambient $SO_2$, warranting future investigation**. However, in general the $SO_2$ sensor contributed the least to model performance, thus this sensor could be replaced with a more relevant sensor, such as NO, in future iterations of the RAMP monitor."

6) All goodness of fit discussions relative to Cross et al., 2017 need to be revised according to the results published in the final accepted version of that manuscript.

**Response:** We have updated our tables and mentions within the manuscript body to be correct and consistent with the final version of the manuscript. We have also removed any mention of the combined testing and training data and updated our comparisons to reflect the new numbers.

7) Additional comments

P11 L15: The figure caption does not indicate this…

- o Figure 4 shows the calibrated RAMP #1 output regressed against the reference monitor concentration for the entire testing period for all three calibration models (LAB, MLR, and RF).

**Response:** Thank you for pointing this out – the figure caption in the manuscript was from a previous iteration of the figure that did not have the regressions. We missed updating it to reflect the new version of the manuscript. The updated caption is below:

**"Figure 4: Example time series and regressions comparing the reference monitor data (black) to statistically average RAMP (RAMP#1) using LAB model (green), multiple linear regression (MLR) model (blue) and random forest (RF) model (pink). The left panel shows only 48 hrs of time series data to illustrate approach; the full evaluations (Table 3) were performed with much larger testing datasets; example regressions from the full data set for RAMP #1 are shown in the right panel."**

P12 L20: The text states that the MAE comparison is against the number of points, but Figure 9 displays this data versus the number of weeks, not number of points.

**Response:** Thank you for pointing this out. This is another example of text that was not fully updated after revising a figure. We have corrected the text to say number of weeks vs number of points. We apologize for the error.

First paragraph of section 2.2 is unnecessarily repetitive
**Response:** Thank you, upon re-reading the paragraph, we agree and have deleted the redundant sentence describing which sensors are in the RAMP monitor.

Modified text (removed sentence in bold and strikethrough):
"The study uses the Real-time Affordable Multi-Pollutant (RAMP) monitor, which was developed in a collaboration between Carnegie Mellon University and SenSevere.  The RAMP uses the following commercially-available electrochemical sensors from Alphasense Ltd: carbon monoxide (CO, Alphasense ID: CO-B41), nitrogen dioxide ($NO_2$, Alphasense ID: NO2-B43F), sulfur dioxide ($SO_2$, Alphasense ID: SO2-B4), and total oxidants ($O_x$, Alphasense ID: Ox-B431). The unit also includes a nondispersive infrared (NDIR) $CO_2$ sensor (SST CO2S-A) which contains built-in T (method: bandgap) and RH (method: capacitive) measurement."

95 sensor measurements (should be 76).

**Response:** As part of our response to your first comment, we have revised the text to be more clear on sensor count:

Modified text in Section 2.2 (additions in bold):
"The experiments involved 95 individual pollutant sensors mounted in 19 unique RAMP monitors. **While the collocation period spanned August 2016-February 2017, many sensors were intermittently deployed for air quality campaigns in Pittsburgh, so the collocation period ranged from 30 days to the full study period, depending on the unit. Additionally, calibrations were not built for sensors for which reference data was below detection limits or if reference monitoring units were malfunctioning, reducing the total number of sensors in this experiment to 73, due to issues with the $SO_2$ and $NO_2$ reference monitors.**"

P7 L6 'beta4' should be 'beta3' according to the formula above
**Response:** Thank you for pointing out this typographical error, it has been corrected to $\beta_3$.

P9 L20 missing 'resolution' following 'temporal'
**Response:** Thank you for pointing out this typographical error, it has been corrected to "a higher temporal resolution"

P11 L13 Figure 2 should read Figure 3
**Response:** Thank you for pointing out this typographical error, it has been corrected to refer to Figure 3 in the text.

P18 L7 missing 'this' - as written: 'demonstrate that degree'
**Response:** Thank you for pointing out this typographical error, we have added the word "this" in the sentence in the Conclusions section.

P20 L30 Levy 2014 reference is the same as Moltchanov et al., 2015 reference.
**Response:** We apologize for the error, it seems as though we had an old version of the reference that was incorrectly imported into our reference software. The references have been merged and we apologize for the mistake.

Figure 2 caption should specify units as 'a.u.' following >255.9
**Response:** Thank you for suggesting this, we have added units to the CO sensor signal in the Figure 2 caption.

Figure 4 (left) – why do the four different pollutant times series all have unique time periods? If environmental parameters impact the sensors differently (RH, T) then it would be important to keep these parameters self-similar across the evaluation framework presented here (even though it's only 48-hours worth, should be the same 48 hours for all sensors).
**Response:** The intent was just as a visual snapshot of a period when there was variation in concentration – we also wanted a period when there was uninterrupted testing data. The ozone data is shifted by about two weeks as at that time the ozone reference monitor was offline. Given

that we are not including T and RH in our plots, we do not feel it is essential to show the same time period, and it would not change the manuscript in any meaningful way.

Figure 7. It's not clear why there are ~ 10 or fewer data points displayed when data from 19 RAMPS are reportedly presented

**Response:** In Figure 7, there are 16 data points for CO and $O_3$, 15 data points for $CO_2$ and 10 data points for $NO_2$, as this is the number of RAMP monitor sesnors included as part of the testing data. We have more clearly addressed this within the manuscript and these changes are included earlier in our response to reviewers (your first comment). In Figure 7 it is occasionally difficult to see all 10-16 data points as there was significant overlap in many of the points (model performance was fairly consistent between RAMP monitors).

Figure 8 caption. 'long periods' is relative. Data displayed is for 15 weeks. Lifetime of the sensors is significantly longer than this (~100-150 weeks). Language should be revised accordingly.
- o The extent to which the model improves over time should be quantified with 95% confidence intervals on the linear fits. By eye, it looks like this confidence interval would include 0.

**Response:** Both within the caption and in the manuscript body, the relative term "long periods" has been replaced with "the study period". Given the relatively new sensors we worked with, the focus of the study was not on temporal degradation which may be where the term "long" is more appropriate, as you mention. Thus, we have modified the manuscript to be more specific that the maximum period is 15 weeks. We have also calculated 95% confidence intervals for the slopes and they do include 0.

Modified text in Section 4.3.1 (additions in bold):

"For all the gas species, the MAE was essentially flat across the RAMP monitors **and the 95% confidence interval on the slope included 0**; RAMP monitors with more testing data did not have substantially higher (worse) MAE, suggesting the RF models are robust **over the study period**."

Table 3. Rather than identifying the number of days of sampling/evaluation – it would be more appropriate to identify the total number of data points used in each case study.
- o Add an extra column that identifies the time resolution – as this is an important factor that drives signal-to-noise and accuracy and precision metrics as well as various end-use cases of interest.

**Response:** We agree that adding either time resolution or number of points would be helpful for others (but both would be redundant). As such we have added a column for time resolution, as we think this would be most helpful for others when considering what sort of model performance is achievable at a given time resolution.

Section 4.4. As written, this section oversimplifies the reality of the situation. When analyzing various lower-cost AQ sensor systems it is important to recognize that the combined hardware and

software configuration impacts the performance metrics, not the software alone. The authors shouldn't gloss over this fact.

**Response:** This point was also mentioned by both Reviewer #1 and Reviewer #2 and we have made changes in the manuscript to address this.

Modified text in introduction (additions in bold):

**"The two primary requirements of low cost sensors for ambient measurement are 1) hardware that is sensitive to ambient pollutant concentrations, and 2) calibration of the sensors. The latter is the focus of this study. A primary challenge of low-cost sensor calibration** is that the sensors are prone to cross-sensitivities with other ambient pollutants (Bart et al., 2014; Cross et al., 2017; Masson et al., 2015b; Mead et al., 2013)"

Modified text in Section 4.4 is below (additions in bold):

For $NO_2$, the performance of 'out-of-the-box' low-cost sensors varied widely and half the sensors in the EuNetAir study (Borrego et al., 2016) reported errors larger than the average ambient concentrations. **While the quality of the baseline gas sensing unit remains critical (in which case no calibration should work), we suggest that advanced calibration models, such as those using machine learning, may be critical for accurate measurements of ambient $NO_2$.**"